Ma *et al. Genome Biology*    (2023) 24:24

METHOD

# BIGKnock: fine-mapping gene-based associations via knockoff analysis of biobank-scale data

Shiyang Ma[1,2], Chen Wang[1], Atlas Khan[3], Linxi Liu[4], James Dalgleish[1], Krzysztof Kiryluk[3], Zihuai He[5,6] and Iuliana Ionita-Laza[1*]

*Correspondence:
ii2135@columbia.edu

[1] Department of Biostatistics, Columbia University, New York, NY, USA
[2] Clinical Research Institute, Shanghai Jiao Tong University School of Medicine, Shanghai, China
[3] Division of Nephrology, Department of Medicine, Vagelos College of Physicians & Surgeons, Columbia University, New York, NY, USA
[4] Department of Statistics, University of Pittsburgh, Pittsburgh, PA, USA
[5] Quantitative Sciences Unit, Department of Medicine, Stanford University, Stanford, CA, USA
[6] Department of Neurology and Neurological Sciences, Stanford University, Stanford, CA, USA

## Abstract

We propose BIGKnock (BIobank-scale Gene-based association test via Knockoffs), a computationally efficient gene-based testing approach for biobank-scale data, that leverages long-range chromatin interaction data, and performs conditional genome-wide testing via knockoffs. BIGKnock can prioritize causal genes over proxy associations at a locus. We apply BIGKnock to the UK Biobank data with 405,296 participants for multiple binary and quantitative traits, and show that relative to conventional gene-based tests, BIGKnock produces smaller sets of significant genes that contain the causal gene(s) with high probability. We further illustrate its ability to pinpoint potential causal genes at ∼ 80% of the associated loci.

**Keywords:** Gene-based associations, Fine-mapping, Knockoff statistics, Algorithmic leveraging, UK Biobank

## Background

Gene-based tests that incorporate regulatory variation from proximal and distal regulatory elements are appealing given that most genetic variants associated with complex traits reside in non-coding regions. Unlike single variant testing which requires follow-up investigations to identify the causal gene(s), gene-based testing that incorporates putative regulatory elements provides a unified test at the gene level. Transcriptome-wide association tests (TWAS) are typical examples of gene-based tests that leverage expression quantitative trait loci (eQTL) data from reference datasets such as GTEx [1]. However, a main challenge is the high false positive rate for such tests caused by confounding due to linkage disequilibrium (LD) and co-regulation. Reducing the number of false positive associations, referred to as fine-mapping, is essential for prioritizing causal genes and for a more mechanistic understanding of genetic associations. Although fine-mapping approaches have been proposed for TWAS [2], these approaches are limited to

eQTLs being present in the reference datasets, and the majority of genetic associations cannot be clearly assigned to existing eQTLs [3–5].

Biobanks with comprehensive genetic and phenotypic data from electronic medical records provide a powerful resource for genomic studies. For example, the UK biobank is comprised of genotype and phenotype data on about 500,000 individuals and millions of genetic variants [6]. In previous work [7] we have proposed a new gene-based test (GeneScan3D) that incorporates genetic variation in proximal and distal regulatory elements (not restricted to eQTLs) and its knockoff version (GeneScan3DKnock) which performs genome-wide conditional tests (on LD) via knockoffs in order to reduce the confounding effect of LD [8]. The idea behind knockoff-based inference is to generate synthetic, noisy copies (knockoffs) of the original genetic variants that resemble the true variants in terms of preserving correlations but are conditionally independent of the phenotype given the true genetic variants. The knockoffs serve as negative controls and help select significant variants while controlling the false discovery rate (FDR). Constructing multiple knockoff genotype features is time consuming and GeneScan3D-Knock cannot be scaled to the large number of individuals and large number of variants available in biobank-scale datasets. In this paper we propose a gene-based test via knockoffs for biobank sized data, BIGKnock. The main ingredient for the improved computational efficiency is the use of a sampling technique based on the empirical statistical leverage scores as an importance sampling distribution [9]. We further take advantage and implement recent developments for linear mixed models [10, 11] that make such models scalable to biobank sized datasets.

We illustrate BIGKnock's performance in terms of power, FDR control and computational cost using simulations. We then demonstrate BIGKnock's ability to prioritize likely causal genes for several binary and continuous traits in the UK biobank data. We illustrate with several loci where BIGKnock prioritizes well known causal genes along with loci with new, plausible causal genes. We also show that the prioritized genes have interesting properties relative to non-significant genes that are consistent with them being putative causal genes. Relative to recent causal gene prioritization methods such as combined SNP-to-gene (cS2G) [12] and Locus-to-gene (L2G) [13] which are based on supervised machine learning methods to integrate various functional features predictive of the causal gene(s) at a locus, and which are therefore dependent on good quality training data and high quality fine-mapping results, our gene-based test avoids these limitations, produces more interpretable results (in terms of *q*-values and FDR control) and naturally restricts false positives due to LD confounding.

## Results

### Overview of BIGKnock

We provide here a brief overview of the proposed gene-based test, BIGKnock. BIG-Knock provides a biobank-scale implementation of GeneScan3DKnock, by implementing a sampling method called algorithmic leveraging, that uses the empirical statistical leverage scores as an importance sampling distribution [9]. BIGKnock computes for each gene a knockoff statistic $W$ that measures the importance of each gene (similar to a *p*-value), and then uses the knockoff filter to detect genes with sufficiently large $W$, i.e., those genes significant at a specified FDR target level [8]. We also compute a q-value for

each gene. A *q*-value is similar to a *p*-value, except that it measures significance in terms of FDR rather than FWER, and already incorporates correction for multiple testing. The details on these specific tests can be found in the Methods section.

We compare the performance of BIGKnock with GeneScan3DKnock in terms of power, FDR control and computational efficiency using simulations. In particular, we show that GeneScan3DKnock incurs substantial computational cost when applied to the UK Biobank data. We illustrate the advantages of the knockoff-based test (BIGKnock) vs. the conventional test (GeneScan3D) using applications to UK Biobank traits.

### Power, FDR and computational cost of BIGKnock in simulations

We perform simulations to evaluate the statistical performance of BIGKnock in terms of power and FDR control, as well as the computational performance for biobank-scale datasets. We sample $n = 10,000$ unrelated individuals from the UK Biobank (European) individuals to evaluate the power and FDR of BIGKnock and GeneScan3DKnock. We randomly select 10 causal genes and 175 noisy genes with gene length $\leq 100$ Kb. For each gene, we include the corresponding GeneHancer and ABC enhancers located within $\pm 100$ Kb of the gene (this restriction is only done in simulations); on average, we include 6.6 enhancers for each selected gene. To avoid FDR inflation due to co-regulation issues, we select genes such that the gene $\pm 100$ Kb region do not overlap for different genes. For each replicate, we set 5% of the variants in the gene $\pm 5$ Kb buffer region (MAF$> 0.01$) to be causal, all located within a randomly selected 10 Kb causal window. Additionally, we also simulate causal variants in enhancers, i.e., we randomly set 5% of the variants in all enhancers for a causal gene to be causal. We generated the continuous/binary traits using the selected causal variants as follows:

- For a continuous trait: $Y_i = X_i + \boldsymbol{G}_i^T \boldsymbol{\beta} + \epsilon_i,$
- For a binary trait: $\text{logit}(P(Y_i = 1)) = X_i + \boldsymbol{G}_i^T \boldsymbol{\beta},$

where $\boldsymbol{G}_i$ denotes the genotypes of causal variants across 10 assumed causal genes and the corresponding enhancers for individual $i = 1, \ldots, n$, and $\boldsymbol{\beta}$ is the vector of the corresponding effect sizes. $X_i \sim N(0, 1)$ is a covariate and $\epsilon_i \sim N(0, 1)$. The case-control ratio for binary phenotype is 1:3. We set the effect size $\beta_j = c|\log_{10}\text{MAF}_j|$. For continuous traits, $c = 0.2$; for binary traits, $c = 0.35$.

The empirical power and FDR are averaged over 200 replicates. The empirical power is defined as the proportion of causal genes being identified; the empirical FDR is defined as the proportion of detected genes that are non-causal. We present results for BIGKnock and GeneScan3DKnock for multiple knockoffs ($M = 5$) and GeneScan3D-BH (the standard Benjamini-Hochberg procedure based on original GeneScan3D *p*-values). We show that both BIGKnock and GeneScan3DKnock control the FDR at the target level, while GeneScan3D-BH has inflated FDR (Fig. 1a, b). BIGKnock has similar power as GeneScan3DKnock. Although GeneScan3D-BH has the highest power, its FDR cannot be controlled (as also shown previously in [7]).

To compare the running time of BIGKnock and GeneScan3DKnock for knockoff generation, we consider varying sample sizes from 1000 to 400,000 in the UK Biobank data. The computing time was evaluated based on 1347 variants in one randomly selected

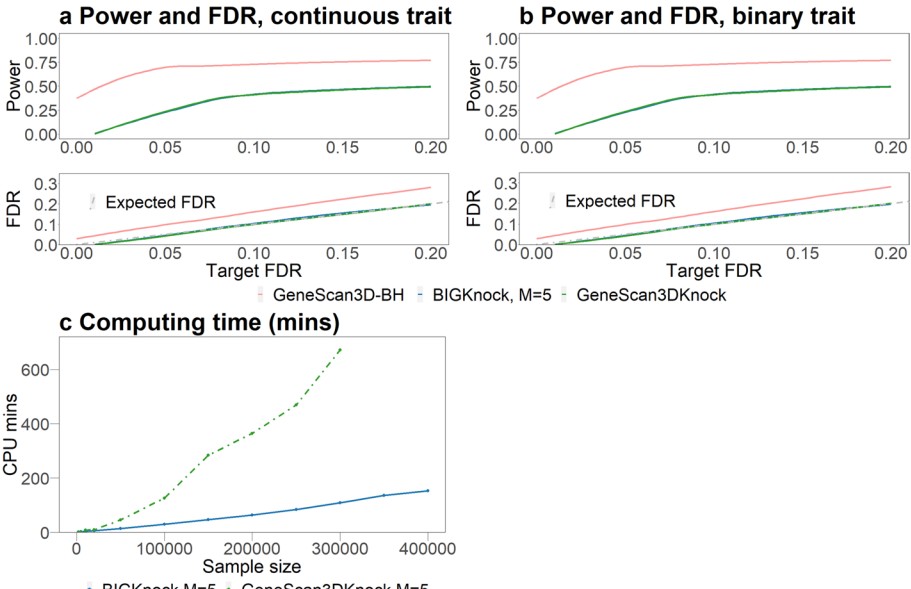

**Fig. 1** Power, FDR and computing time comparisons for different methods. **a** and **b** Power and FDR comparisons between GeneScan3D-BH, GeneScan3DKnock and BIGKnock ($M$=5 knockoffs) for continuous and binary traits. **c** Computing time for different methods to generate knockoffs: GeneScan3DKnock and BIGKnock (with shrinkage algorithmic leveraging). The computing time were evaluated based on a gene with 1347 variants, varying the sample size from 1000 to 400,000

gene. BIGKnock with shrinkage algorithmic leveraging only takes 2.5 CPU hours to run for 400,000 individuals (Fig. 1c). However, GeneScan3DKnock is not able to finish computation in less than 12 CPU hours for $n > 300,000$, so the computing time is truncated at $n = 300,000$. The computing time was evaluated on a single CPU (Intel Xeon CPU E5-2630 @ 2.30GHz).

### Applications to UK Biobank: binary traits

We applied BIGKnock to nine binary traits in the UK Biobank, including hypertension, coronary artery disease (CAD), asthma, type 2 diabetes (T2D), hypothyroidism, hyperlipidemia, skin cancer, varicose veins, and inguinal hernia (See Additional file 1: Table S1 for sample size information). Note that we have previously [7] compared the performance of the original knockoff-based test, GeneScan3DKnock, and GeneScan3D with other commonly-used tests including STAAR-O [14] and MAGMA/H-MAGMA [15, 16], and have shown improved power and FDR control relative to these existing methods. Therefore, in these applications we directly compare BIGKnock with the conventional test (GeneScan3D) to illustrate the advantages of the knockoff-based testing approach. We use a Bonferroni adjusted threshold of $2.5 \times 10^{-6}$ for GeneScan3D and an FDR threshold of 0.01 or 0.05 (depending on the size of the study) for BIGKnock. For nine binary traits we consider here, we identify 2295 gene-trait associations for GeneScan3D and 1555 associations for BIGKnock (Additional file 2: Supplementary Tables 6-14). Among the 2295 significant associations under GeneScan3D, only 1349 (58.8%) are significant under BIGKnock, despite the more liberal (FDR) threshold used by BIGKnock. This reduction in the number of significant associations can be, in part,

attributed to removal of some false positive associations due to the LD adjustment within the knockoff framework, consistent with simulation results showing inflated FDR for GeneScan3D.

We use the significant GWAS SNPs ($p < 5 \times 10^{-8}$) to define 1Mb loci centered at the most significant SNP. For each gene-based test (GeneScan3D and BIGKnock), we count the number of loci that contain at least one significant gene for each test respectively. In terms of the number of significant loci, GeneScan3D and BIGKnock show similar results, with most of the significant loci shared between GeneScan3D and BIGKnock (Additional file 1: Table S2). However, one of our main interests in employing the knockoff framework is to filter out false positive genes that appear in the conventional GeneScan3D test. We therefore consider shared loci that contain at least one significant gene for both GeneScan3D and BIGKnock, and compare the number of significant genes identified by the two methods at such loci. The knockoff test discovers a smaller number of significant genes than GeneScan3D despite the more liberal FDR threshold (Fig. 2, Additional file 1: Fig. S1). We provide further evidence below that BIGKnock, by conditioning on nearby variants, can prioritize genes more likely to be causal.

### BIGKnock can prioritize putative causal genes at significant loci

We demonstrate that significant genes detected by BIGKnock tend to be enriched among genes nearest to the lead GWAS SNP at significant loci, the class of genes most likely to be the causal genes [17, 18]. We first perform the enrichment analysis (Methods) based on 257 BIGKnock significant loci for multiple binary traits. Knockoff significant genes are 4.5-fold (range 2.4–10.5 for nine binary traits) more likely to be the nearest gene relative to the rest of the genes at a locus (Fig. 3a). Similar results hold when we restrict the analyses to 245 loci shared between BIGKnock and GeneScan3D (4.7-fold with range 2.4–9.1, Additional file 1: Fig. S2(a)).

Next, we focus on several loci where the knockoff-based test can prioritize only a few genes at a locus relative to the conventional GeneScan3D test (Table 1), and there is compelling literature support for a mechanistic role of the selected gene(s) in the pathogenesis of the corresponding traits.

### ALDH2 (aldehyde dehydrogenase 2) and coronary artery disease

We illustrate first in detail the association between *ALDH2* and coronary artery disease. Although GeneScan3D identifies 12 significant genes at this locus, BIGKnock identifies only three of them as significant including *ALDH2*, *BRAP*, and *ATXN2* (Fig. 4a). The additional associations detected by the conventional GeneScan3D test are likely due to LD between variants in those genes and putative causal variants in the *ATXN2-ALDH2* neighborhood. *ALDH2* is expressed across many tissues in GTEx but is most abundant in the liver and adipose tissues (Fig. 4c). The role of *ALDH2* in cardiovascular disease is well-documented in the literature [19]. The *ALDH2* Glu504lys polymorphism is widely considered as a risk factor for the development of coronary artery disease, especially in Asian populations [20–22]. Furthermore, mitochondrial ALDH2 has emerged as a key enzyme for removal of ethanol-derived acetaldehyde, and has been shown to play a role in inflammation regulation and macrophages accumulation [23]. Epidemiological studies in humans carrying an inactivating mutation in *ALDH2*, combined with genetic and

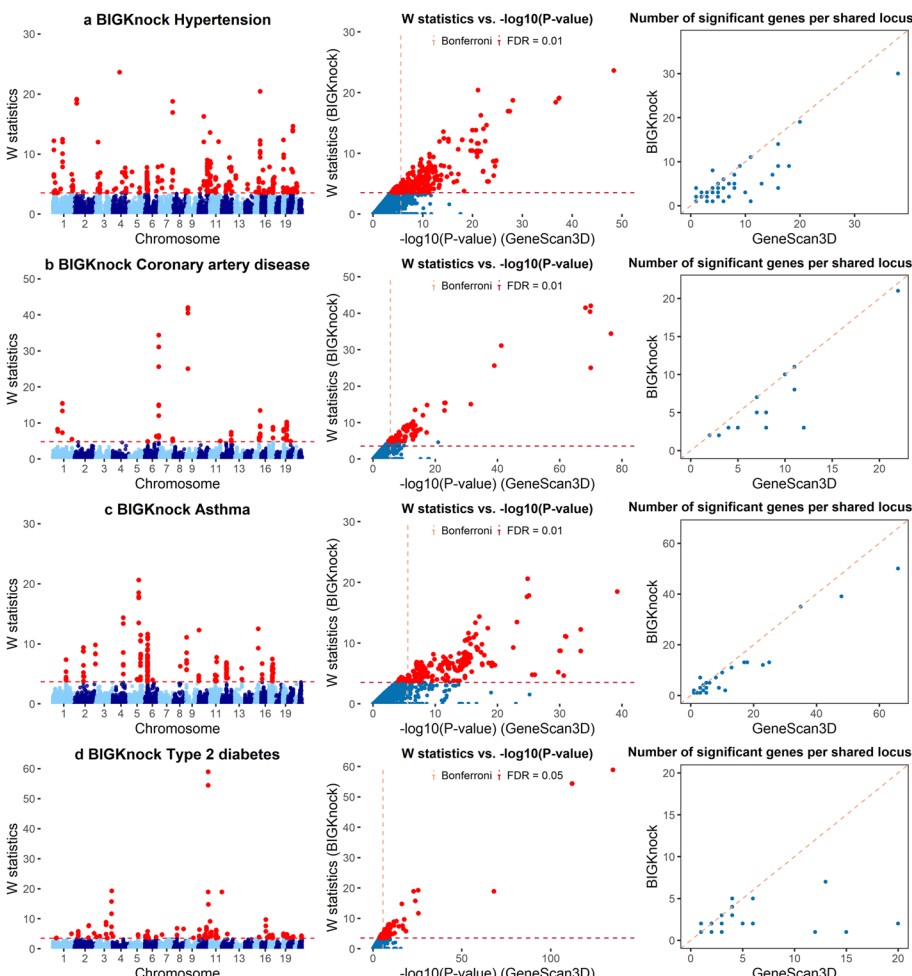

**Fig. 2** Applications to UK Biobank binary traits (1). **a**–**d**, Manhattan plots for BIGKnock, Scatter plot of *W* knockoff statistics (BIGKnock) vs. −log$_{10}$(*p* value) (GeneScan3D), and Scatter plot of the number of significant genes per locus between conventional GeneScan3D and BIGKnock are shown for **a** hypertension, **b** coronary artery disease, **c** asthma, and **d** type 2 diabetes. The dashed lines in the left and middle panels show the significance thresholds defined by Bonferroni correction (for *p*-values) and by false discovery rate (FDR; for W statistic)

pharmacological studies in animal models, have implicated *ALDH2* in the development and prognosis of coronary heart disease, hypertension, type 2 diabetes, and stroke, and suggest *ALDH2* as an important target for generating new treatments for heart diseases [24].

### Additional loci with strong literature support

*NGFR* (nerve growth factor receptor) and asthma (Fig. 5a): Nerve growth factor has been implicated in both the immune and neuronal components of allergic asthma pathogenesis. Furthermore, the nerve growth factor (*NGF*) targeting treatment may be an important therapy for antigen-induced airway hyper responsiveness via attenuation of airway innervation and inflammation in asthma [25].

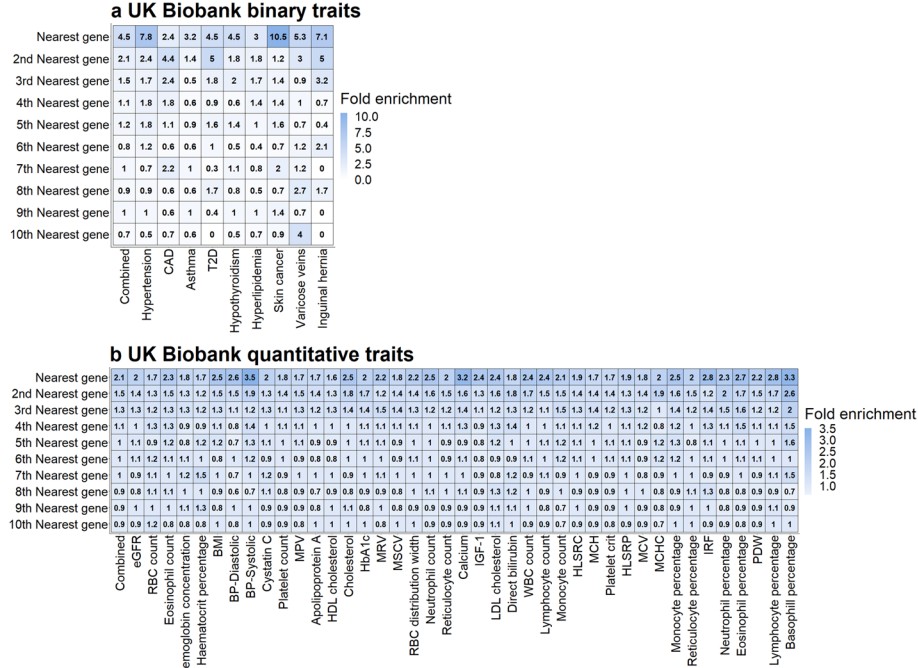

**Fig. 3** Enrichment of BIGKnock significant genes among genes closest to the lead GWAS variant at BIGKnock significant loci. Enrichment of BIGKnock significant genes for **a** the nine combined binary traits and each binary trait separately; and **b** the 41 combined quantitative traits and each quantitative trait separately

*AGPAT1* (1-acylglycerol-3-phosphate O-acyltransferase 1) and type 2 diabetes (Fig. 5b): *AGPAT1* is a metabolism (lipid biosynthesis) gene and plays important functions in the physiology of multiple organ systems. In particular, Agpat1-deficient mouse developed widespread disturbances of metabolism including low body weight and low plasma glucose levels [26]. Furthermore, Agpat1 mouse knockout has low circulating glucose and increased urine glucose and urine microalbumin (International Mouse Phenotyping Consortium).

*MARCHF5* (membrane-associated RING-CH-type finger 5) and type 2 diabetes (Fig. 5c): *MARCHF5* is a PPARγ target gene that influences mitochondrial and cellular metabolism in adipocytes [27]. These functions likely alter the utilization of lipid, which subsequently impacts glucose metabolism.

*IL2RA* (interleukin 2 receptor subunit alpha) and hypothyroidism (Fig. 5d): *IL2RA* is involved in the regulation of T-cell function and has been related to autoimmune thyroid disease (AITD) [28].

*CD69* (CD69 molecule) and hypothyroidism (Fig. 5e): Levels of CD69+ regulatory lymphocytes are increased in autoimmune thyroid disorder patients [29].

*HMGCR* (3-hydroxy-3-methylglutaryl-CoA reductase) and hyperlipidemia (Fig. 5f): *HMGCR* has been identified as one of the therapeutic targets of hypercholesterolemia. It is a major point of control in cholesterol homeostasis and *HMGCR* and *PCSK9* inhibitors have been widely used to treat hypercholesterolemia in clinical settings [30].

*APOA5* (apolipoprotein A5) and hyperlipidemia (Fig. 5g): There are multiple lines of evidence linking *APOA5* and hyperlipidemia. For example, the *APOA5* gene was found

**Table 1** Selected loci for binary and quantitative traits. The number of significant genes per locus for GeneScan3D, BIGKnock, and BIGKnock significant genes are shown. The putative causal gene is shown in boldface font

| BIGKnock locus-trait | position (hg19) | # GeneScan3D | # BIGKnock | BIGKnock genes |
|---|---|---|---|---|
| BRAP-CAD | 12: 111,986,818–112,986,818 | 12 | 3 | **ALDH2, ATXN2, BRAP** |
| UBE2Z-Asthma | 17: 46,948,346–47,948,346 | 11 | 2 | **NGFR, UBE2Z** |
| AGPAT1-T2D | 6: 32,126,272–33,126,272 | 12 | 1 | **AGPAT1** |
| CPEB3-T2D | 10: 93,966,910–94,966,910 | 6 | 2 | **MARCHF5, CPEB3** |
| IL2RA-Hypothyroidism | 10: 5,606,266–6,606,266 | 8 | 2 | *GDI2*, **IL2RA** |
| CD69-Hypothyroidism | 12: 94,22,652–10,422,652 | 7 | 2 | **CD69, CLEC2B** |
| HMGCR-Hyperlipidemia | 5: 74,105,220–75,105,220 | 7 | 1 | **HMGCR** |
| BUD13-Hyperlipidemia | 11: 116,148,917–117,148,917 | 12 | 3 | **APOA5, BUD13, ZNF259** |
| ASGR1-Cholesterol | 17: 6,569,412–7,569,412 | 43 | 2 | **ASGR1, CD68** |
| SLC39A8-BP-Diastolic | 4: 103,269,304–104,269,304 | 6 | 1 | **SLC39A8** |
| DBH-BP-Diastolic | 9: 135,649,709–136,649,709 | 6 | 3 | *ADAMTS13*, **DBH, SARDH** |
| ANGPTL4-Cholesterol | 19: 7,951,937–8,951,937 | 9 | 1 | **ANGPTL4** |
| RAB11A-Neutrophil count | 15: 65,544,465–66,544,465 | 8 | 1 | **RAB11A** |
| ZHX3-Calcium | 20: 39,455,078–40,455,078 | 6 | 1 | **ZHX3** |
| PPARG-LDL cholesterol | 3: 11,739,931–12,739,931 | 6 | 1 | **PPARG** |
| POLDIP2-LDL cholesterol | 17: 26,194,861–27,194,861 | 22 | 3 | **POLDIP2, SLC13A2, TMEM199** |
| E2F4-RBC count | 16: 66,229,250–67,229,250 | 18 | 3 | **E2F4, FAM96B, KIAA0895L** |
| ENSA-WBC count | 1: 150,095,537–151,095,537 | 26 | 2 | *ENSA*, **MCL1** |
| E2F2-Lymphocyte count | 1: 23,019,508–24,019,508 | 11 | 1 | **E2F2** |
| KCTD17-Hematocrit percentage | 22: 36,962,936–37,962,936 | 11 | 3 | *KCTD17*, *MPST*, **TMPRSS6** |
| SAMD7-MCH | 3: 169,029,895–170,029,895 | 10 | 3 | *ACTRT3*, *LRRC31*, **SAMD7** |
| ITGA4-Monocyte percentage | 2: 182,392,917–183,392,917 | 6 | 3 | *CERKL*, **ITGA4, NEUROD1** |

associated with familial combined hyperlipidemia and dyslipidemia in large dutch families [31], and in an italian population [32].

### *Effector BIGKnock genes*

We further restrict the list of BIGKnock significant genes by identifying those that coincide with the closest gene (among all genes) to the top significant GWAS SNP at a locus. Among 257 significant BIGKnock loci across nine binary traits, we identify 178 (69%) such loci. We call these genes effector BIGKnock genes. For loci that do not have effector BIGKnock genes, 35 loci have only one BIGKnock significant gene. Therefore, we

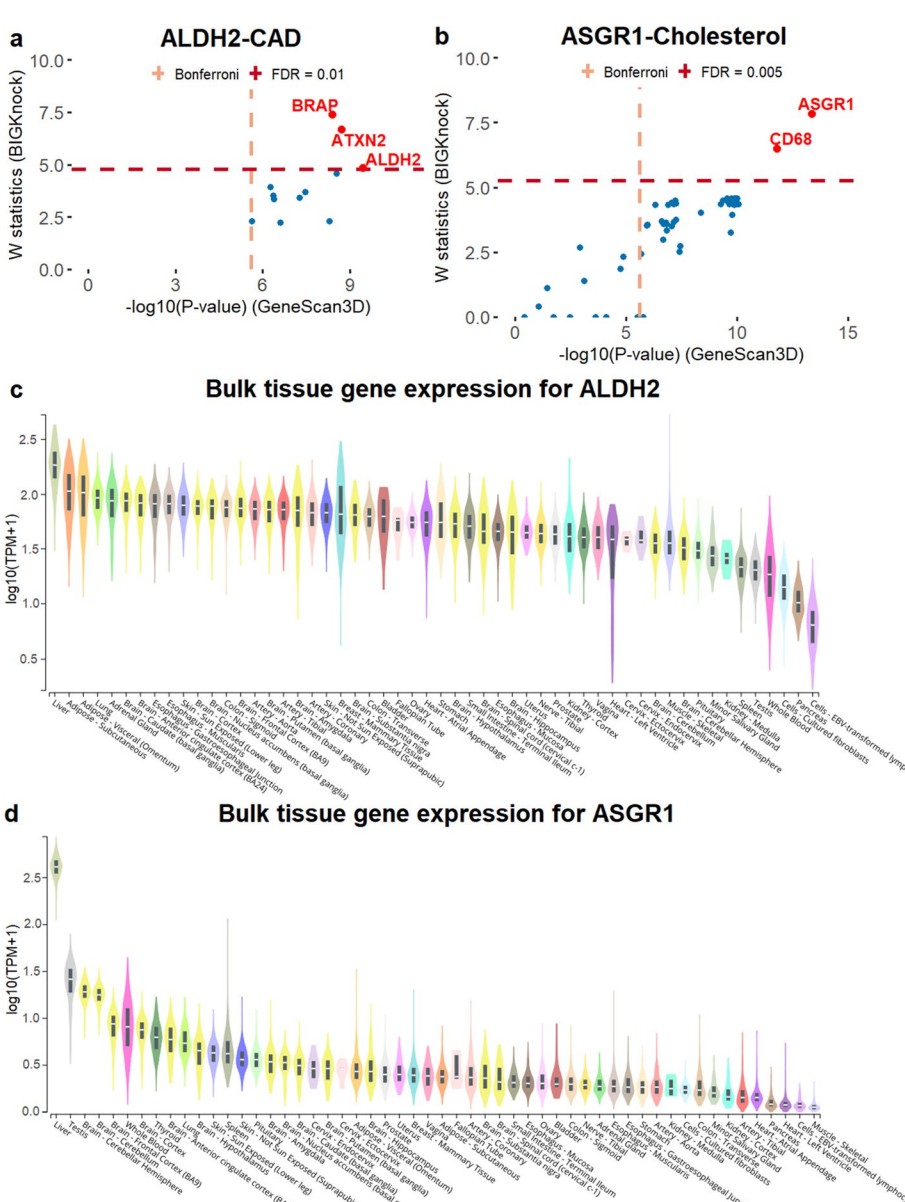

**Fig. 4** ALDH2-CAD and ASGR1-Cholesterol loci. **a** Scatter plot of *W* knockoff statistics (BIGKnock) vs. $-\log_{10}(p$ value) (GeneScan3D) for the ALDH2-CAD locus, **b** Scatter plot of *W* knockoff statistics (BIGKnock) vs. $-\log_{10}(p$ value) (GeneScan3D) for the ASGR1-Cholesterol locus, **c** GTEx gene expression across tissues for *ALDH2*, and **d** GTEx gene expression across tissues for *ASGR1*

prioritize potentially causal genes at 213 (83%) loci (Additional file 2: Supplementary Table 57).

### Mouse phenotype enrichment analyses

Using ToppFun [33] we have tested whether the effector BIGKnock genes are enriched in sets of genes associated with mouse phenotypes. The mouse phenotype data are extracted from the Mammalian Phenotype Ontology, and consists of mouse genes that

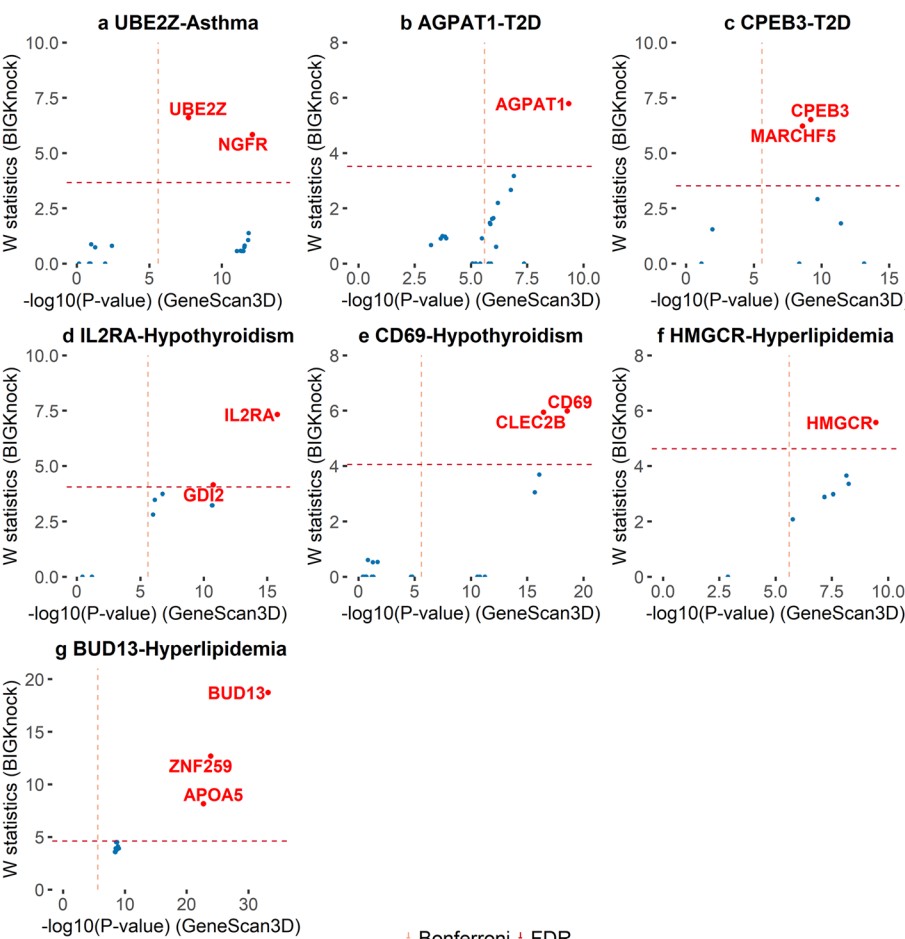

**Fig. 5** Putative causal genes at selected loci for UK Biobank binary traits. Scatter plots of *W* knockoff statistics (BIGKnock) vs. $-\log_{10}(p$ value) (GeneScan3D) for selected loci of **a** asthma, **b–c** type 2 diabetes (T2D), **d–e** hypothyroidism, and **f–g** hyperlipidemia. Loci are named according to the most significant gene in BIGKnock. The dashed lines show the significance thresholds defined by Bonferroni correction (for *p*-values) and by false discovery rate (FDR; for W statistic).

cause phenotypes in genetically engineered or mutagenesis experiments. Effector BIG-Knock genes are enriched in gene sets corresponding to relevant mouse phenotypes (Additional file 1: Fig. S3). For example, among the most significantly enriched phenotypes were abnormal circulating insulin levels, and abnormal glucose tolerance for Type 2 diabetes, abnormal systemic arterial blood pressure for hypertension, abnormal CD4-positive, alpha-beta T cell physiology and abnormal T-helper 2 physiology for asthma, abnormal hepatobiliary system physiology for coronary artery disease, decreased cholesterol level and decreased circulating cholesterol level for hyperlipidemia and abnormal skin pigmentation for skin cancer.

### Applications to UK Biobank: quantitative traits

We have also applied BIGKnock to 41 quantitative traits in the UK Biobank (Additional file 1: Table S3).

For quantitative traits we use more stringent FDR thresholds (0.001 or 0.005) relative to binary traits due to the much larger sample sizes and consequently large number of significant findings. For these 41 quantitative traits, we identify 125,246 gene-trait associations for GeneScan3D and 80,917 associations for BIGKnock (Additional file 2: Supplementary Tables 15-55). Among 125,246 associations significant under GeneScan3D, only 78,614 (62.8%) are significant under BIGKnock similar to the binary traits above.

We report the number of significant loci/genes per trait in Additional file 1: Table S4. As with the binary traits, for most of the significant shared loci, BIGKnock can reduce the number of significant associations despite the more liberal (FDR) thresholds being used (Additional file 1: Figs. S4-S11).

### BIGKnock can prioritize putative causal genes at significant loci

As with binary traits, we demonstrate that significant genes detected by BIGKnock tend to be enriched among genes nearest to the lead GWAS SNP at significant loci. We first perform the enrichment analysis (Methods) on 13,548 BIGKnock significant loci for multiple quantitative traits. In particular, knockoff significant genes are 2.1-fold (range 1.6–3.5 for individual traits) more likely to be the nearest gene relative to the rest of the genes at a locus (Fig. 3b). When we restrict the analyses to 13,224 loci shared between BIGKnock and GeneScan3D, similar enrichment can be observed (Additional file 1: Fig. S2(b)).

Next, we focus on several loci where the knockoff-based test can prioritize few genes at a locus relative to GeneScan3D (Table 1), and there is compelling literature support for a mechanistic role of the selected genes in the pathogenesis of the corresponding traits.

### ASGR1 (asialoglycoprotein receptor 1) and cholesterol

We illustrate first in detail the example of *ASGR1* and cholesterol. At the 1 Mb locus containing *ASGR1*, BIGKnock prioritizes two genes including *ASGR1* among 43 genes significant using the conventional GeneScan3D test (Fig. 4b). Most of the GeneScan3D associations are due to gene-enhancer links for two enhancers (Additional file 1: Fig. S12). Specifically, 18 associations are due to variants in an enhancer GH17F007167 just upstream of gene *ASGR1*, and when accounting for LD with nearby variants, BIGKnock no longer detects them as significant. Furthermore, additional associations that are removed by BIGKnock are 12 genes linked to ABC enhancer chr17:7,144,929–7,146,587 (hg19) downstream of gene *ASGR1*, and 6 genes linked to 4 other enhancers (Additional file 2: Supplementary Table 56). Therefore, at this locus, BIGKnock is able to prioritize two genes by adjusting for linkage disequilibrium in the region. *ASGR1* is also highly expressed in liver (Fig. 4d). The role of *ASGR1* in the control of non-HDL cholesterol levels and in regulation of the endogenous levels of at least some asialoglycoproteins has been established [34]. Specifically, Nioi et al. [34] have identified rare loss-of-function variants in *ASGR1* that are associated with lowering of non-HDL cholesterol levels and a reduced risk of coronary artery disease. Recent mechanistic studies also support a role of *ASGR1* in cholesterol. For example, ASGR1-deficient pigs show lower levels of non-HDL cholesterol and less atherosclerotic lesions than that of controls, therefore

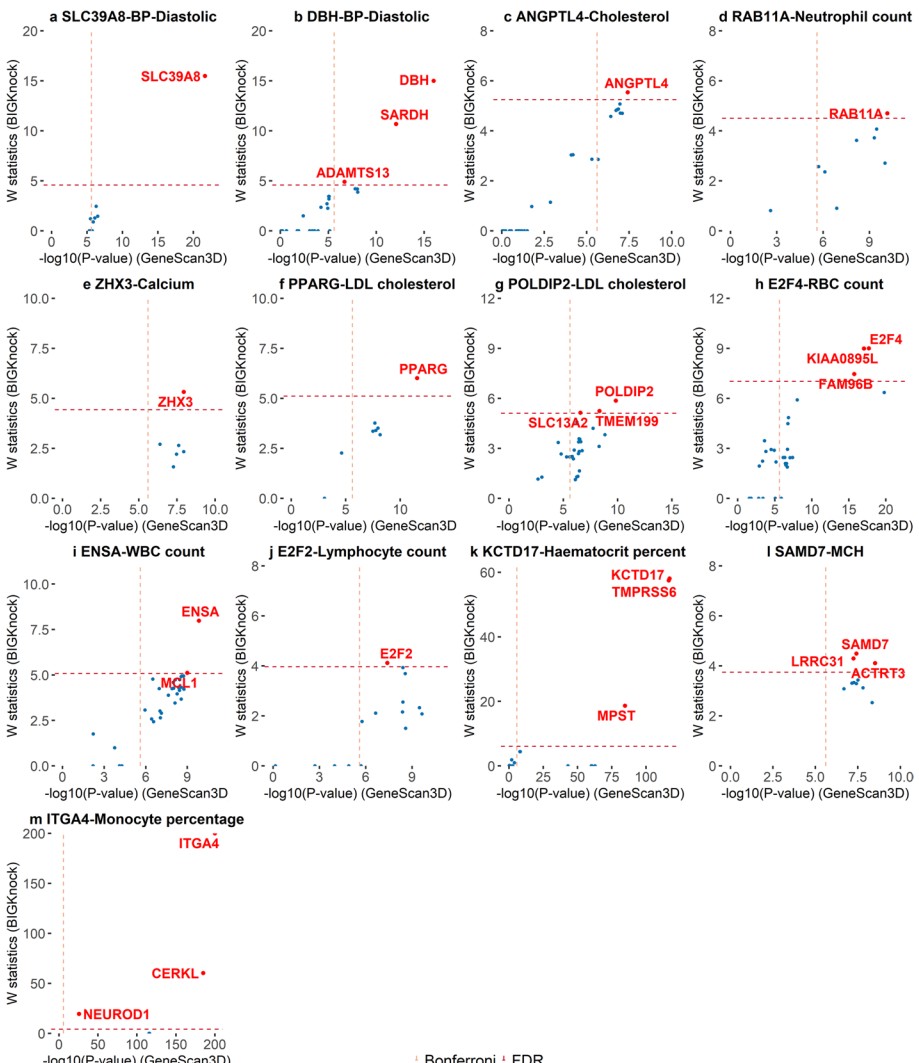

**Fig. 6** Putative causal genes at selected loci for UK Biobank quantitative traits. Scatter plots of *W* knockoff statistics (BIGKnock) vs. $-\log_{10}(p$ value) (GeneScan3D) for selected loci of **a**–**b** BP-diastolic, **c** cholesterol, **d** neutrophil count, **e** calcium, **f**–**g** LDL cholesterol, **h** RBC count, **i** WBC count, **j** lymphocyte count, **k** hematocrit percentage, **l** MCH, and **m** monocyte percentage. Loci are named according to the most significant gene in BIGKnock. The dashed lines show the significance thresholds defined by Bonferroni correction (for *p*-values) and by false discovery rate (FDR; for W statistic)

targeting ASGR1 might be an effective strategy to reduce hypercholesterolemia and atherosclerosis [35].

### *Additional loci with strong literature support*

*SLC39A8* (solute carrier family 39 member 8) and diastolic blood pressure (Fig. 6a): Slc39a8 deletion in mice results in increased nitric oxide (NO) production, decreased blood pressure, and protection against high-salt-induced hypertension, while homozygosity of the *SLC39A8* loss-of-function variant in humans is associated with increased NO, providing a plausible explanation for the association of *SLC39A8* with blood pressure [36, 37].

*DBH* (dopamine beta-hydroxylase) and diastolic blood pressure (Fig. 6b): Dbh($-/-$) mice had a low heart rate, were severely hypotensive, and displayed an attenuated circadian blood pressure rhythm [38].

*ANGPTL4* (angiopoietin-like protein 4) and cholesterol (Fig. 6c): *ANGPTL4* was uncovered as a novel modulator of plasma lipoprotein metabolism. In 24-h fasted mice, Angptl4 overexpression increased plasma triglycerides (TG) by 24-fold, which was attributable to elevated VLDL-, IDL/LDL-, and HDL-TG content [39].

*RAB11A* (ras-related protein Rab-11A) and neutrophil counts and neutrophil percentage (Fig. 6d): In mice challenged with endotoxin, intratracheal instillation of Rab11a-depleted macrophages reduced neutrophil count in bronchoalveolar lavage fluid, increased the number of macrophages containing apoptotic neutrophils, and prevented inflammatory lung injury [40].

*ZHX3* (zinc fingers and homeoboxes 3) and calcium (Fig. 6e): Zhx3-KO mice have increased bone mineral density (International Mouse Phenotyping Consortium), and ZHX3 may be useful as an early osteogenic differentiation marker [41].

*PPARγ* (peroxisome proliferator- activated receptor gamma) and LDL cholesterol (Fig. 6f): PPARγ regulates fatty acid storage and glucose metabolism. The genes activated by PPARγ stimulate lipid uptake and adipogenesis by fat cells. PPARγ plays a regulatory role in the first steps of the reverse-cholesterol-transport pathway through the activation of ABCA1-mediated cholesterol efflux in human macrophages [42].

*POLDIP2* (polymerase delta-interacting protein 2) and LDL cholesterol (Fig. 6g): Poldip2 was shown to increase Nox4 enzymatic activity by 3-fold and to positively regulates basal reactive oxygen species production in vascular smooth muscle cells [43]. The authors suggest that Poldip2 may be a novel therapeutic target for vascular pathologies with a significant vascular smooth muscle cell migratory component, such as restenosis and atherosclerosis.

*E2F4* (E2F transcription factor 4) and red blood cell (erythrocyte) count (Fig. 6h): *E2F4* is essential for normal erythrocyte maturation and neonatal viability, which makes a major contribution to the control of erythrocyte development [44]. Besides, *E2F4* can regulate fetal erythropoiesis through the promotion of cellular proliferation [45].

*MCL1* (myeloid cell leukemia-1) and white blood cell (leukocyte) count (Fig. 6i): Induced myeloid leukemia cell differentiation protein Mcl-1 is a protein encoded by the *MCL1* gene and is essential for the survival of neutrophils (polymorphonuclear leukocytes) [46]. Furthermore, a novel class of Mcl-1 inhibitors has the potential to be developed for the treatment of acute myeloid leukemia [47].

*E2F2* (E2F transcription factor 2) and lymphocyte count (Fig. 6j): E2Fs are important regulators of proliferation, differentiation, and apoptosis. Mutations in *E2F2* in mice cause enhanced T lymphocyte proliferation, leading to the development of autoimmunity [48]. Furthermore, the combined loss of *E2F1* and *E2F2* was shown to have profound effects on hematopoietic cell proliferation and differentiation, as well as increased tumorigenesis and decreased lymphocyte tolerance [49].

*TMPRSS6* (transmembrane serine protease 6) and hematocrit percentage (Fig. 6k): *TMPRSS6* is a well-known red blood cell traits related gene [50] and also one of the effector genes identified by [51]. *TMPRSS6* is essential for normal systemic iron homeostasis

in humans and mutations in *TMPRSS6* may cause iron-refractory iron deficiency anemia [52].

*SAMD7* (sterile alpha motif domain containing 7) and mean corpuscular hemoglobin (Fig. 6l): The hepatocyte-specific *SAMD7* knockout mice show decreased iron and hemoglobin concentration [53]. *SAMD7* deficiency may decrease iron and hemoglobin through hepcidin up-regulation.

*ITGA4* (integrin subunit alpha 4) and monocyte percentage (Fig. 6m): *ITGA4* has been recently associated with inflammatory bowel disease [54, 55]. Additionally, an eQTL for *ITGA4* is strongly associated with monocyte counts [56].

### Effector BIGKnock genes

We further restrict the list of BIGKnock significant genes by identifying those that coincide with the closest gene (among all genes) to the top significant GWAS SNP at a locus. Among 13,548 significant loci across 41 quantitative traits, we identify 8530 (63%) such loci. Furthermore, for significant loci that do not contain effector BIGKnock genes, an additional 1799 (13%) loci have only one gene significant under BIGKnock. Therefore, using the BIGKnock significant genes we can prioritize potentially causal genes for 76% of the loci (Additional file 2: Supplementary Table 58).

### Mouse phenotype enrichment analyses

Using ToppFun [33], we have tested whether the effector BIGKnock genes are enriched in sets of genes associated with mouse phenotypes (Additional file 1: Figs. S14-S17). Effector BIGKnock genes are enriched in gene sets corresponding to relevant mouse phenotypes. For example, among the most significantly enriched phenotypes were abnormal systemic arterial blood pressure for BP-diastolic, abnormal erythroid lineage cell morphology and abnormal erythrocyte morphology for RBC count, abnormal calcium ion homeostasis for calcium, abnormal circulating LDL cholesterol level for LDL cholesterol, abnormal circulating HDL cholesterol level for HDL cholesterol, abnormal circulating hormone level for IGF-1, abnormal blood cell morphology/development for MPV, abnormal hemoglobin for HbA1c, increased circulating bilirubin level for direct bilirubin, increased lymphocyte cell number for lymphocyte count, abnormal mean corpuscular volume for MCH and MCV, abnormal cellular hemoglobin content for MCHC, abnormal mononuclear cell morphology for monocyte percentage and abnormal T cell morphology for Lymphocyte percentage.

### Putative causal gene identification using BIGKnock

#### BIGKnock can prioritize putative effector genes in Backman et al. [51]

We use data on putative effector genes identified in a recent study by Backman et al. [51] using rare-variant exome-wide association studies in 454,787 participants in the UK Biobank study. Specifically, Backman et al. first identify common variants independently associated with each trait (i.e., GWAS sentinel variant), which are then included as additional covariates for Burden association tests with rare variants focusing on pLOF (including stop-gain, frameshift, stop-loss, start-loss and essential splice variants) and deleterious missense variants with a minor allele frequency (MAF) of up to 1%. Overall, 168 significant genes (with 584 gene-trait associations for 216 traits) adjusting for

GWAS signals (with Burden $p$-values $\leq 2.18 \times 10^{-11}$) and that are nearest to the GWAS sentinel variant are defined as the likely effector genes [51]. Here we consider the 201 effector gene-trait associations corresponding to 43 binary and quantitative traits considered in our analyses (Additional file 2: Supplementary Table 59). We identify 194 effector gene associations that are significant under GeneScan3D with 173 (89%) also significant under BIGKnock. Note that this is a significantly higher retention rate for effector gene associations than the expected rate (62.7%, 79,963 BIGKnock significant genes out of 127,541 GeneScan3D significant genes; Fig. 7) based on all genes significant under GeneScan3D for binary and quantitative traits (two-sided $p= 4.4 \times 10^{-14}$), and supports the claim that BIGKnock retains the truly causal genes while removing many of the false associations due to LD. Several examples include *ASGR1* and *SH2B3* and cholesterol, *APOB* and apolipoprotein A, *TMPRSS6* and hematocrit percentage, and *SH2B3* and WBC count. *ANGPTL4* was also prioritized by BIGKnock for cholesterol, and identified as effector gene for HDL cholesterol (Additional file 1: Table S5 and Fig. S13).

In addition, Backman et al. [51] identified 564 genes associated with traits using rare variant association tests focusing as above on pLOF and deleterious missense variants with a MAF of up to 1%. Among 233 genes that correspond to 37 quantitative traits considered in our analyses (Additional file 2: Supplementary Table 60), we identify 186 GeneScan3D significant genes with 163 (88%) being significant under BIGKnock. Again, this is a significantly higher proportion than expected based on all GeneScan3D associations (two-sided $p=3.6 \times 10^{-12}$). Several example include *DBH* associated with BP-Diastolic, *SLC5A3* associated with Cystatin C, *POLE* associated with MRV, *E2F8* associated with MCV, *SIGIRR* associated with Lymphocyte count and *EPB41* associated with HbA1c (Additional file 2: Supplementary Table 61).

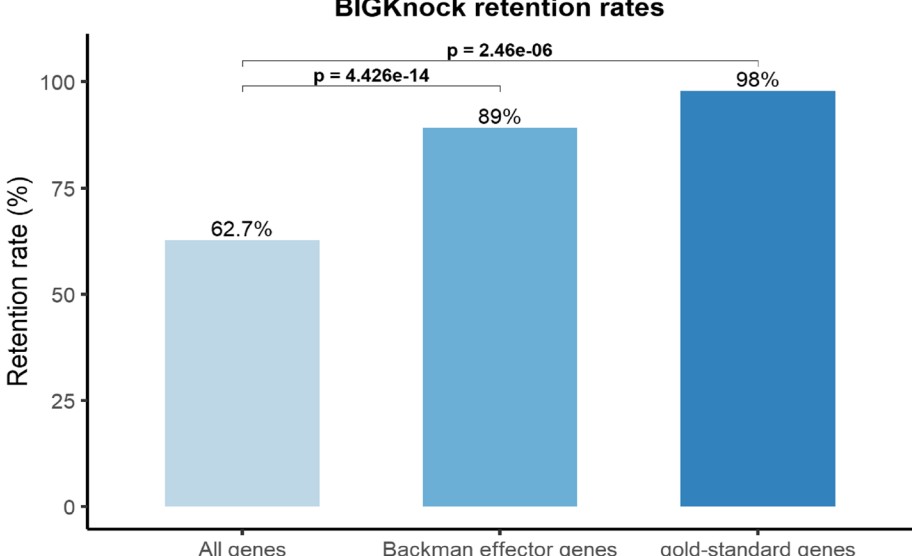

**Fig. 7** BIGKnock retention rates for all genes, Backman effector genes and gold-standard genes, with two-sided *p*-values. The retention rates are computed as the proportions of BIGKnock significant genes among the GeneScan3D significant genes using three different gene sets: all genes and two gene sets enriched for causal genes (Backman effector genes, and gold-standard genes)

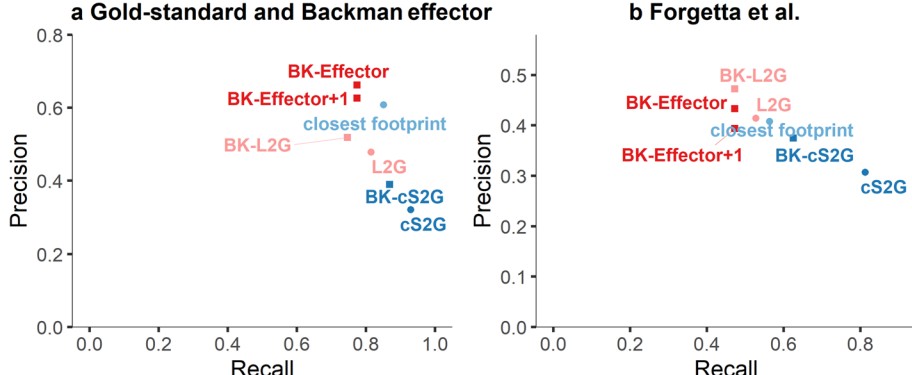

**Fig. 8** Comparisons of different locus-to-gene prioritization methods. Precision vs. recall is shown for several representative methods including closest footprint, cS2G, L2G, BIGKnock Effector genes (BK-Effector), BIGKnock Effector genes and genes at BIGKnock significant loci with only one significant gene (BK-Effector+1), as well as combination of BIGKnock and cS2G (BK-cS2G) and L2G (BK-L2G). **a** Gold standard and Backman effector dataset including 221 positive genes at BIGKnock significant loci. The negative genes include 3255 genes located within the 1Mb loci containing the 221 positive genes; **b** Forgetta et al. gene set including 55 positive genes at BIGKnock significant loci. The negative genes include 860 genes located at 1Mb loci containing the 55 positive genes

Another recent study using whole-exome sequencing data on 200,337 UK Biobank participants and focused on cardiometabolic traits has also performed exome-wide rare variant analyses with rare (pLOF and deleterious missense) variants [57]. Restricting to the traits included in our analyses (hypertension, hypothyroidism, type 2 diabetes, BMI, HDL, LDL, and IGF-1) and the 25 gene-trait associations with $q$-value< 0.05 in [57], we find that 18 of them are significant in GeneScan3D, of which 16 (89%) are significant in BIGKnock (two-sided p= $4 \times 10^{-2}$; Additional file 2: Supplementary Table 62).

### Comparisons with other locus-to-gene linking methods on gold-standard gene sets

We have compared the accuracy of effector BIGKnock genes to other methods to prioritize putative causal genes at GWAS loci, including the closest gene footprint to the top GWAS SNP as well as more recent methods such as combined SNP-to-gene (cS2G) [12] and locus-to-gene (L2G) [13], using two gold-standard gene sets from the literature. Note that BIGKnock is mainly a gene-based test that uses individual level data and therefore different in nature to these existing gene prioritization methods; nonetheless, it is interesting to compare its performance with such methods in terms of ability to prioritize causal genes at loci of interest.

Specifically, we first consider 49 expert-curated genes with high confidence [13] (note that 45 of these genes are significant in GeneScan3D, with 44/45 being BIGKnock significant, i.e., retention rate 98%; Fig. 7), as well as 201 effector genes identified using rare pLOF variants in [51]. For all our analyses here we focus on 221 gene-trait associations overlapping loci that are significant using the BIGKnock test. As control genes we consider the remaining genes at those loci for a total of 3255 genes. For cS2G, we further restrict to a subset of 114 positive genes and 1651 control genes for 18 traits analyzed both here and in [12]. We compare methods in terms of precision and recall, where precision for a method is computed as the fraction of positive genes among the genes prioritized by that method, and recall is computed as the fraction of positive genes

prioritized by that method (Fig. 8a). BIGKnock effector genes have the highest precision among all methods considered, i.e., 0.66; the recall is also high (0.77). By comparison, cS2G achieves a higher recall (0.93 for cS2G score > 0.5) with a greatly reduced precision (0.32). Closest gene footprint has a slightly higher recall (0.85) relative to effector genes, but reduced precision (0.61), while L2G has similar recall (0.81) but lower precision (0.48) relative to effector genes. Furthermore, combining BIGKnock with other scores (such as cS2G and L2G) generally leads to improved precision over the individual cS2G and L2G scores (Additional file 2: Supplementary Table 63).

We consider a second set of stringently defined positive genes, including Mendelian disease genes and drug targets as described in Forgetta et al. [58]. Specifically, we consider 208 genes that corresponded to traits type 2 diabetes, hyperthyroidism, BP-systolic, BP-diastolic, LDL-cholesterol, calcium, direct bilirubin, and red blood cell count considered in our analyses. We focus on 55 genes residing at BIGKnock significant loci. As control genes we consider all genes at these 1Mb loci for a total of 860 genes. For cS2G, we further focus on a subset of 48 gold-standard genes and 743 control genes for 6 traits analyzed both here and in [12] (calcium and direct bilirubin do not have cS2G gene scores). BIGKnock effector genes have the highest precision among all individual methods considered, i.e., 0.43; the recall is also relatively high (Fig. 8b, 0.47). By comparison, cS2G achieves a higher recall (0.81) but at a greatly reduced precision (0.31). Closest gene footprint has higher recall (0.56), but slightly lower precision (0.41). L2G has slightly higher recall (0.53) and lower precision (0.41). Furthermore, combining BIG-Knock with other scores (such as cS2G and L2G) generally leads to improved precision over the individual methods (Additional file 2: Supplementary Table 64).

Finally, we have compared BIGKnock with L2G and cS2G for several known causal genes, including *ASGR1*-Cholesterol, *ANGPTL4*-Cholesterol and *ALDH2*-CAD, as previously discussed (Additional file 1: Fig. S18). For *ASGR1*, all three methods identify *ASGR1* with high scores; however, cS2G identifies four such genes at the locus. For *ANGPTL4*, only BIGKnock and cS2G identify it among high scoring genes. However, cS2G identifies three other genes with similar high score at this locus. For *ALDH2*, only BIGKnock and L2G identify it among the highest scoring gene; however, L2G identifies six such genes at this locus. Results for other putative causal genes and loci highlighted before are similar (Additional file 1: Figs. S19-S21).

### *Characteristics of prioritized genes*

We have focused here on prioritizing genes at ∼ 80% loci that have either effector genes, i.e., the gene closest to the most significant GWAS SNP is significant using the BIG-Knock test, or loci where BIGKnock prioritizes only one gene. We show that these genes have certain interesting properties: (1) have significantly higher pLI scores, (2) they have significantly longer CDS (Coding DNA Sequence), and (3) higher LOF mutation rates than genes that are never selected by BIGKnock across a variety of binary and quantitative traits considered here (Additional file 1: Fig. S22). This latter result is consistent with previous studies that showed that highly conserved genes (including putative disease causing genes) have, rather counterintuitively, higher mutation rates. Specifically, Michaelson et al. [59] showed that hypermutability is correlated with highly conserved

sequence using whole genome sequencing data. Although the exact mechanisms underlying this relationship are not known, one possible explanation is that these genes, on account of their essential nature, are highly transcribed and consequently more susceptible to transcription-mediated mutagenic events.

## Discussion

A main limitation of gene-based tests when incorporating putative regulatory variants, such as eQTLs or variants residing in regulatory elements such as promoters and enhancers, is the potentially high false positive rate due to LD confounding and co-regulation. We propose here a gene-based test for biobank-scale data that reduces the confounding effect due to LD and can prioritize putative causal genes at GWAS significant loci. For co-regulation, e.g., when a causal enhancer may regulate multiple genes and hence will be included in the testing for multiple genes, the knockoff-based framework cannot help. The proposed test goes beyond state-of-the-art gene-based tests by allowing integration of a wider class of regulatory variants than eQTLs, and by performing conditional analysis (on LD), thereby adjusting for LD.

We show that BIGKnock reduces the number of significant associations at a locus relative to conventional tests despite a more liberal FDR adjustment, and retains with high probability ($\sim$ 90%) the likely causal genes as shown using the effector and rare variant association results in [51]. Furthermore, between 63 and 69% of loci with BIGKnock significant genes have the closest gene to the top GWAS SNP at the locus being significant under BIGKnock (Additional file 2: Supplementary Tables 57 and 58). In addition to such effector BIGKnock genes, BIGKnock also prioritizes genes that are not necessarily nearest to the top GWAS SNP. Overall, approximately 80% of loci have one single gene prioritized based on significant genes detected by BIGKnock.

BIGKnock is complementary to other locus-to-gene strategies in the literature that are based on supervised machine learning models and fine-mapping results. BIGKnock prioritizes causal genes via a formal gene-based test that limits confounding due to LD relative to existing tests in the literature. Therefore BIGKnock is less functionally informed relative to existing locus-to-gene strategies, and therefore less affected by potential biases in existing training datasets. Combining significant genes in BIGKnock with other functionally informed causal gene prioritization methods is a promising avenue for increasing performance. We show that relative to other causal gene prioritization approaches, the proposed method has improved precision while achieving high recall, which is important in this setting due to costly follow-up functional studies.

Although it is a challenging task to prove that the prioritized genes from any method are indeed causal, we show multiple lines of evidence from mouse phenotype data, curated gold-standard gene lists, mutation rate data and supporting literature that BIGKnock is helpful in identifying putative causal genes including several examples with known causal links in the literature such as *ASGR1* and *ANGPTL4* and cholesterol, and *ALDH2* and coronary artery disease. These prioritized genes can serve as good candidates for further functional studies.

We have implemented BIGKnock in a computationally efficient R package that can be applied generally to the analysis of biobank scale data.

## Conclusions

BIGKnock is a powerful and computationally efficient gene-based test that leverages long-range chromatin interaction data, is applicable to biobank-scale data, and performs conditional testing genome-wide via model-X knockoffs. Unlike conventional tests, it helps reduce confounding due to LD and hence prioritizes causal associations over those induced by LD. As a method to prioritize causal genes at a GWAS locus, it has advantages over existing supervised machine learning models in that it produces interpretable results (*q*-values) and is not dependent on possibly biased training data. Furthermore, it can be combined with existing prioritization scores to improve their performance. BIGKnock has been implemented in a computationally efficient package and can be applied widely to biobank-scale data analyses.

## Methods

### Overview of GeneScan3D and its knockoff-based extension GeneScan3DKnock

We first describe the details of a previously developed gene-based test (GeneScan3D) that incorporates noncoding variants using long-range chromatin data [7]. Assume there are *n* samples with *p* variants in a gene plus buffer region as well as the corresponding regulatory elements. For *i*-th individual, we denote $Y_i$ as the phenotype, $G_i$ as the $p \times 1$ genotype vector and $X_i$ as the $d \times 1$ covariate vector including an intercept. We are interested in testing for association between the phenotype and the *p* variants, while adjusting for covariates. For unrelated individuals, we consider the generalized linear model (GLM):

$$g(\mu_i) = X_i^T \alpha + G_i^T \beta,$$

where $\mu_i$ is the conditional mean of phenotype $Y_i$ conditional on covariates, $\alpha$ is a $d \times 1$ vector of regression coefficients for *d* covariates (including an intercept) and $\beta$ is a $p \times 1$ vector of regression coefficients for *p* variants.

We scan the gene plus buffer region ($\pm$ 5 Kb) using *L* 1D windows with sizes 1–5–10 Kb, then construct 3D windows by adding one enhancer to each 1D window. For each gene, we focus on GeneHancer and ABC enhancers [60, 61] that are outside the gene plus buffer region, containing at least 2 variants and with length between 0.5 Kb and 10 Kb. In the ABC model [61], we only incorporate predicted ABC enhancers with ABC scores $\geq 0.02$ for 5 human cell types and tissues, i.e., K562, GM12878, NCCIT, LNCAP, hepatocytes.

Assuming *R* enhancers for a gene, then we construct $L \times R$ 3D windows. For each 3D window, we conduct (i) Burden and Sequence Kernel Association Test (SKAT) tests [62] for all common variants (MAF > 0.01) within the window, using equal weights; and (ii) single variant score tests for individual common variants with MAF > 0.01. The Cauchy combination method [63] is applied to combine *p*-values from the above tests within each 3D window. Finally, we compute the GeneScan3D *p*-value by combining $L \times R$ 3D window's *p*-values using Cauchy combination method.

### GeneScan3DKnock: knockoff-based extension

By incorporating distal regulatory elements, gene-based tests can leverage noncoding genetic variation to improve power of gene-based tests. However, due to linkage disequilibrium (LD) and/or co-regulation of multiple genes by the same regulatory element, many of the significant genes may be false positives. Hence, we previously developed GeneScan-3DKnock [7], a knockoff-based test to attenuate the confounding effect of LD and prioritize putative causal genes with controlled false discovery rate (FDR). Note that co-regulation is still a problem and cannot be addressed by the proposed approach.

To generate multiple knockoff genotypes, we consider the general sequential conditional independent tuples approach [8, 64, 65]. Specifically, we sequentially sample $\tilde{G}_j^1, \ldots, \tilde{G}_j^M$ independently from $\mathcal{L}(G_j | G_{-j}, \tilde{G}_{1 \ldots j-1}^1, \ldots, \tilde{G}_{1 \ldots j-1}^M)$, where $M$ is the number of knockoffs. Note that we can leverage the approximate block structure for LD in the genome to only condition on variants in a neighborhood $B_j$ of the current variant $j$. The knockoff genotypes are exchangeable with the original genotypes $G$, and lead to guaranteed FDR control. With the assumption that genotypes can be approximately modeled by a multivariate normal distribution, we consider a computational efficient auto-regressive model to estimate:

$$\hat{G}_j = \hat{\alpha} + \sum_{k \neq j, k \in B_j} \hat{\beta}_k G_k + \sum_{m=1}^{M} \sum_{k \leq j-1, k \in B_j} \hat{\gamma}_k^m \tilde{G}_k^m. \tag{1}$$

We estimate the coefficients using the least squares method:

$$(\hat{\alpha}, \hat{\boldsymbol{\beta}}, \hat{\boldsymbol{\gamma}}) = [\text{cov}(\mathbf{1}, \boldsymbol{G}_{B_j}, \tilde{\boldsymbol{G}}_{B_j})]^{-1} (\mathbf{1}, \boldsymbol{G}_{B_j}, \tilde{\boldsymbol{G}}_{B_j})^T \boldsymbol{G}_j,$$

where

$$cov(\mathbf{1}, \mathbf{G}_{\mathbf{B}_j}, \tilde{G}_{B_j}) = \begin{pmatrix} 1 & 0 & 0 \\ 0 & G_{B_j}^T G_{B_j} & G_{B_j}^T \tilde{G}_{B_j} \\ 0 & \tilde{G}_{B_j}^T \mathbf{G}_{\mathbf{B}_j} & \tilde{G}_{\mathbf{B}_j}^T \tilde{G}_{B_j} \end{pmatrix}.$$

$\mathbf{G}_{\mathbf{B}_j}$ and $\tilde{G}_{B_j}$ correspond to the original $\mathbf{G}_{\mathbf{k}}$ and previously generated knockoff variants with $k \neq j, k \in B_j$. By calculating the residual $\hat{\boldsymbol{\epsilon}}_j = \boldsymbol{G}_j - \hat{\boldsymbol{G}}_j$ and its $M$ permutation, the knockoff features $\tilde{\boldsymbol{G}}_j^m = \hat{\boldsymbol{G}}_j + \hat{\boldsymbol{\epsilon}}_j^{*m}$ are obtained. Note that we can replace the sample covariance matrix above by a low rank approximation based on spectral decomposition.

After generating multiple knockoffs, we conduct the proposed gene-based test on the original genotype and knockoff genotypes for each gene. The feature statistic for each gene $G$ is then defined as

$$W_G = (T_G - \text{median } T_{\tilde{G}}^m) I_{T_G \geq \max_{1 \leq m \leq M} T_{\tilde{G}}^m},$$

where $T_G = -\log_{10}(p_G)$ and $T_{\tilde{G}}^m = -\log_{10}(p_{\tilde{G}}^m)$ are the importance score for gene $G$ in original genotype and knockoff cohort, and $I$ is an indicator function. We compute the threshold $\tau$ for FDR control at a certain level $q$:

$$\tau = \min \left\{ t > 0 : \frac{\frac{1}{M} + \frac{1}{M} \#\{G : \kappa_G \geq 1, \tau_G \geq t\}}{\#\{G : \kappa_G = 0, \tau_G \geq t\}} \leq q \right\},$$

where $\kappa_G = \text{argmax}_{0 \leq m \leq M} T_{\tilde{G}}^m$ (note that $T_{\tilde{G}}^0 = T_G$) and $\tau_G = T_G - \text{median } T_{\tilde{G}}^m$. Finally, we select as significant those genes with $W_G \geq \tau$.

### q-value

We additionally compute the corresponding $q$-value for a gene, $q_G$. The q-value already incorporates correction for multiple testing, and is defined as the minimum FDR that can be attained when all tests showing evidence against the null hypothesis at least as strong as the current one are declared as significant. In particular, we define the q-value for a gene $G$ with feature statistic $W_G > 0$ as

$$q_G = \min_{t \leq W_G} \frac{\frac{1}{M} + \frac{1}{M} \#\{G : \kappa_G \geq 1, \tau_G \geq t\}}{\#\{G : \kappa_G = 0, \tau_G \geq t\}},$$

where $\frac{\frac{1}{M} + \frac{1}{M} \#\{G : \kappa_G \geq 1, \tau_G \geq t\}}{\#\{G : \kappa_G = 0, \tau_G \geq t\}}$ is an estimate of the proportion of false discoveries for multiple knockoffs if we were to select all genes with $\kappa_G = 0, \tau_G \geq t$ (with $t > 0$). For genes with feature statistic $W_G = 0$ (i.e., $\kappa_G \geq 1$), we set $q_G = 1$ and never select those genes.

### Shrinkage leveraging algorithm for knockoffs generation

The computational cost of knockoff generation for multiple knockoffs is substantial for biobank-scale data with hundreds of thousands of samples and millions of genetic variants. One commonly used strategy to improve the computational efficiency when dealing with large-scale datasets is sampling. Here, to reduce the computational time and make the test scalable to biobank sized datasets, we employ the shrinkage leveraging (SL) algorithm [9, 66]. The SL algorithm is a sampling technique based on the empirical statistical leverage scores as an importance sampling distribution. Specifically, the method samples rows of the genotype data matrix to reduce the data size before performing computations on the subproblem.

We draw $r = 10n^{1/3} \log n$ subsamples from $n$ samples with importance sampling probabilities:

$$\pi_i = 0.5\pi_i^{\text{Lev}} + 0.5\pi_i^{\text{Unif}}, i = 1, \ldots, n,$$

where $\pi_i^{\text{Unif}} = 1/n$ follows uniform distribution and $\pi_i^{\text{Lev}} = \sum_{j=1}^{p} U_{ij}^2 / \sum_{i=1}^{n} \sum_{j=1}^{p} U_{ij}^2$, $U$ is the orthogonal singular vectors of $(1, G_{B_j}, \tilde{G}_{B_j})$. We then form a weighted linear regression model (1) with weights $w_i = 1/(r\sqrt{\pi_i})$ for $r$ subsamples and compute the least square estimates $(\hat{\alpha}^{\text{SL}}, \hat{\beta}^{\text{SL}}, \hat{\gamma}^{\text{SL}})$. Finally, we generate the knockoff features for $n$ samples using $(\hat{\alpha}^{\text{SL}}, \hat{\beta}^{\text{SL}}, \hat{\gamma}^{\text{SL}})$. In summary, we select a subset of "informative" samples to estimate intermediate parameters used for knockoff generation and thus improve the computational efficiency of the knockoff framework. To efficiently store the knockoff genotypes, we use the Genomic Data Structure compressed files based on *gdsfmt* R package [67].

*Note.* To filter out highly-correlated variants, we apply hierarchical clustering before doing the knockoff generation. We compute correlations for all pairs of variants in regions containing the gene plus buffer region ($\pm 100$ Kb neighborhood) and enhancers ($\pm 50$ Kb neighborhood). Variants with correlation $\geq 0.75$ are clustered together and one representative variant is selected for each cluster. Specifically, if a cluster contains variants inside the

gene plus buffer/enhancer region, we randomly select one representative variant from inside the gene plus buffer/enhancer region. Otherwise, we randomly select one variant as representative.

### Generalized linear mixed effects model for related samples

Unlike GeneScan3DKnock which does not account for sample relatedness, in BIGKnock we incorporate generalized linear mixed-effects model (GLMM) adapted to biobank-scale data. Specifically, we assume:

$$g(\mu_i) = X_i\,\boldsymbol{\alpha} + G_i\,\boldsymbol{\beta} + b_i,$$

where the random effect $\boldsymbol{b} = (b_1, \ldots b_n)^T \sim \mathrm{MVN}(\boldsymbol{0}, \tau\boldsymbol{\psi})$ and $\boldsymbol{\psi}$ is the $n \times n$ genetic relationship matrix (GRM).

Following SAIGE-Gene [11], we consider three steps for the UK Biobank data. In step 1 we construct the sparse GRM $\boldsymbol{\psi}_S$ with cutoff $0.125$ for $n = 405,296$ British samples using 106,256 pruned markers. In step 2 we fit the null GLMM for binary and quantitative traits. Both steps are using the existing software implementation in SAIGE/SAIGE-Gene [10, 11]. In step 3 we perform the gene-based test for each gene using the fitted values $\hat{\mu}$ and estimated variance ratio $\hat{r}$ obtained in step 2. Note that due to the light sample relatedness of UK Biobank data, one can use the sparse GRM to fit null GLMM and estimate variance ratio, which is much more computationally efficient than using the dense GRM [11].

To fit GLMM under the null hypothesis $H_0 : \boldsymbol{\beta} = \boldsymbol{0}$ in a computationally efficient way, SAIGE uses the preconditioned conjugate gradient method [68] that allows calculating the log quasi-likelihood and average information without taking the inverse of $n \times n$ matrix. Specifically, SAIGE maximizes the log quasi-likelihood using the average information restricted maximum likelihood algorithm (AI-REML) [69] to iteratively estimate $(\hat{\boldsymbol{\alpha}}, \hat{\boldsymbol{b}}, \hat{\phi}, \hat{\tau})$ (note that the dispersion parameter $\hat{\phi} = 1$ for binary traits). Denote $\hat{\Sigma} = \hat{W}^{-1} + \hat{\tau}\boldsymbol{\psi}$, where $\hat{W} = \hat{\phi}^{-1}I$ for quantitative traits and $\hat{W} = \mathrm{diag}(\hat{\mu}_1(1 - \hat{\mu}_1), \ldots \hat{\mu}_n(1 - \hat{\mu}_n))$ for binary traits. Denote the covariate-adjusted genotype matrix as $\tilde{G} = G - X(X^T WX)^{-1}X^T WG$ and the projection matrix $\hat{P} = \hat{\Sigma}^{-1} - \hat{\Sigma}^{-1}X(X^T\hat{\Sigma}^{-1}X)^{-1}X^T\hat{\Sigma}^{-1}$.

After fitting the null GLMM, we obtain the variance ratio $\hat{r} = \tilde{\boldsymbol{g}}^T\hat{P}\tilde{\boldsymbol{g}}/\tilde{\boldsymbol{g}}^T\hat{P}_S\tilde{\boldsymbol{g}}$ where $\tilde{\boldsymbol{g}}$ is the covariate-adjusted single variant genotype vector, $\hat{P}_S = \hat{\Sigma}_S^{-1} - \hat{\Sigma}_S^{-1}X(X^T\hat{\Sigma}_S^{-1}X)^{-1}X^T\hat{\Sigma}_S^{-1}$ and $\hat{\Sigma}_S = \hat{W}^{-1} + \hat{\tau}\boldsymbol{\psi}_S$. The variance ratio, which is estimated using a set of 30 randomly selected variants and shown to be approximately constant for all variants [10], is used to calibrate the score test statistics and variance-covariance matrix of gene-based tests for GLMM.

For the single variant score test in GeneScan3D, $S_j = \sum_{i=1}^{n} \tilde{G}_{ij}(Y_i - \hat{\mu}_i)/\hat{\phi}$. We consider the variance-adjusted test statistic:

$$T_j^{\mathrm{adj}} = \frac{S_j}{\sqrt{\tilde{\boldsymbol{g}}_j^T\hat{P}\tilde{\boldsymbol{g}}_j}},$$

where $\tilde{\boldsymbol{g}}_j$ is the covariate-adjusted genotype vector of $j$th variant. The approximation of $\mathrm{var}(S_j) = \tilde{\boldsymbol{g}}_j^T\hat{P}\tilde{\boldsymbol{g}}_j = \hat{r}\tilde{\boldsymbol{g}}_j^T\hat{P}_S\tilde{\boldsymbol{g}}_j \approx \hat{r}\tilde{\boldsymbol{g}}_j^T\hat{\Sigma}_S^{-1}\tilde{\boldsymbol{g}}_j$ and the score test $p$-value can be computed based on $S_j^2/\mathrm{var}(S_j) \sim \chi_{\mathrm{df}=1}^2$.

The Burden and SKAT test statistics in GeneScan3D can be written as:

$$Q_{\text{Burden}} = \left( \sum_{j=1}^{p} w_j S_j \right)^2, \quad Q_{\text{SKAT}} = \sum_{j=1}^{p} w_j^2 S_j^2,$$

where $w_j$ is the weight of each variant. The joint null distribution of $\boldsymbol{S} = (S_1, \ldots S_p)$ follows a multivariate normal distribution with mean $\boldsymbol{0}$ and covariance matrix $\tilde{G}^T \hat{P} \tilde{G} = G^T \hat{\Sigma}^{-1} G - (G^T \hat{\Sigma}^{-1} X)(X^T \hat{\Sigma}^{-1} X)^{-1}(X^T \hat{\Sigma}^{-1} G) = G^T \hat{P} G$. We adjust the covariance matrix for GLMM as $K = \hat{r} G^T \hat{P}_S G$. Since both $\hat{\Sigma}$ and $G$ are sparse matrices, $K$ can be calculated by using the sparse LU decomposition (solve function in R) for each 3D window. Then the Burden $p$-value is obtained from a scaled chi-square distribution $\tilde{\lambda}_B \chi_1^2$, where $\tilde{\lambda}_B = (w_1, \ldots, w_p) K (w_1, \ldots, w_p)^T$. The SKAT $p$-value is obtained from a mixture of chi-square distribution $\sum_{j=1}^{p} \tilde{\lambda}_{S_j} \chi_1^2$ using Davies method [70], where $\tilde{\lambda}_{S_j}$ are the eigenvalues of $\text{diag}(w_1, \ldots, w_p) K \text{diag}(w_1, \ldots, w_p)$.

### Saddlepoint approximation for gene-based test

One challenge for binary traits in biobanks is the possibility of highly unbalanced case:control ratios. In such cases we implement the saddlepoint approximation (SPA) to recalibrate the score test statistics for gene-based testing [71, 72]. Specifically, under case-control imbalance, the distribution of score statistics $\boldsymbol{S} = (S_1, \ldots S_p)$ is skewed, in which case one needs to adjust the covariance matrix $K$ using SPA. As in [11, 72], we first compute the $p$-values of single-variant score test by SPA $\tilde{p}_j$, then the SPA-adjusted variance $\tilde{v}_j = S_j^2 / Q(1 - \tilde{p}_j)$, where $Q$ is the quantile function of $\chi_1^2$. The adjusted covariance matrix $\tilde{K} = \sqrt{\tilde{V}} K \sqrt{\tilde{V}}$, where $\tilde{V} = \text{diag}(\tilde{v}_1 / \hat{v}_1, \ldots, \tilde{v}_p / \hat{v}_p)$ and $\hat{v}_j = K[j, j]$ is the estimated variance of $S_j$. The adjusted covariance matrix $\tilde{K}$ is used to compute the SPA gene-based $p$-values of SKAT and Burden.

### UK Biobank data analyses

The UK Biobank data contains data on 488,377 individuals. All individuals underwent genome-wide genotyping with UK Biobank Axiom array from Affymetrix and UK BiLEVE Axiom arrays ($\sim$ 825,000 markers). Genotype imputation was carried out using a 1000 Genomes reference panel with IMPUTE4 software [6]. We apply several quality-control filters, keeping only variants with MAF$>$ 0.01 imputed with high confidence ($R^2 \geq 0.8$). This resulted in 9,233,477 imputed variants that were available for the analyses. We restrict our analyses to 405,296 participants (218,068 females and 187,228 males) with British ancestry. We adjust for covariates including sex, age, age$^2$, age $\times$ sex and 5 principal components. For principal component analysis, we used a set of common genotypes (MAF$>$ 0.01) pruned using the following command in PLINK –indep-pairwise 500 50 0.05 with 35,226 pruned variants using FlashPCA [73]. A total of 17,753 genes with gene length $<$ 500 kb and with at least 2 variants in the gene plus buffer region were tested. The details on the traits analyzed are given in Tables S1, S3.

We use 106,256 pruned genotyped markers to construct the sparse GRM with relatedness coefficient cutoff $\geq$ 0.125, then fit null GLMMs for several binary and quantitative traits using SAIGE [10, 11]. The 106,256 markers were pruned from the UK Biobank genotype data using PLINK with pairwise LD threshold $r^2 \leq 0.05$, MAF$>$ 0.01, 95% genotyping rate, window size of 500 bp and step size 50 bp. Based on the sparse GRM, there are 21,397 related pairs among the 405,296 participants, including 8 duplicate twins

(kinship coefficient >0.354), 8275 1$^{st}$-degree relatives (kinship coefficient between 0.177 and 0.354) and 13,114 2$^{nd}$-degree relatives (kinship coefficient ≤ 0.177) [74].

### Enrichment of BIGKnock associations among genes closest to lead GWAS SNPs

We consider the significant loci for different UK Biobank binary and quantitative traits. We use the SAIGE summary statistics from the existing UK Biobank studies for binary traits (https://pheweb.org/UKB-SAIGE/) and the GWAS summary statistics for UK Biobank quantitative traits were obtained from the Neale Lab (http://www.nealelab.is/uk-biobank). For each significant locus, all genes within the locus are ranked according to the distance to the lead GWAS variant. The enrichment is then defined as the ratio of the proportion of BIGKnock significant genes that are ranked k-th and the proportion of the remaining genes at the locus that are ranked k-th, where $k = 1, \ldots, 10$.

#### *Gene footprint*

We compute the distance from the lead GWAS variant to gene footprint. The gene footprint can be any position between the start and end positions of the gene [75]. Specifically, if the lead variant is inside the gene, then the distance from the lead variant to the gene footprint is 0. If the variant is outside the gene, the distance to the gene footprint is the smallest distance from the lead variant to any position in the gene (start or end position).

### Locus-to-gene scores

#### *L2G*

We selected GWAS analyses from the OpenTarget Genetics Portal [13] to match the 50 traits tested by BIGKnock. For the nine binary traits we use summary statistics from SAIGE [10]. For ten quantitative traits (Apolipoprotein A, Calcium, Cholesterol, Cystatin C, Direct bilirubin, eGFR, Glycated hemoglobin HbA1c, HDL cholesterol, IGF-1, and LDL direct) we use summary statistics from [76]. We use summary statistics from the Neale lab UKB GWAS round 2 results for BMI, Systolic blood pressure, and Diastolic blood pressure. For the remaining 28 quantitative traits, we use the summary statistics from [77]. OpenTarget used the "locus-to-gene" (L2G) model to prioritize likely causal genes at each GWAS locus detected by these studies. An L2G score is derived from gene distance, molecular QTL colocalization, chromatin interaction, and pathogenicity to quantify the causal probability of a gene. We downloaded the L2G scores and selected the gene with the highest L2G score for each GWAS locus for the 50 traits.

#### *cS2G*

The combined SNP-to-gene strategy (cS2G) [12] includes seven SNP-to-gene (S2G) linking strategies such as Exon, Promoter, two fine-mapped cis-eQTL strategies, EpiMap enhancer-gene linking, Activity-By-Contact, and Cicero. A cS2G score is computed for a SNP and a gene as a linear combination of linking scores from different S2G strategies, and the optimal weights are estimated to maximize the recall under a constraint of precision $\geq 0.75$ with non-trait-specific training critical gene set. cS2G was applied to fine-mapping results of 49 UK Biobank diseases and traits; a cS2G score > 0.5 was used to identify high-confidence SNP-gene-disease triplets. In our analyses, we considered

the cS2G predicted target genes of fine-mapping results for 22 UKBB traits: hypertension, CAD (cardiovascular disease in cS2G), asthma, T2D, hypothyroidism, RBC count, eosinophil count, BMI, BP-diastolic, BP-systolic, platelet count, MPV, HDL cholesterol, cholesterol, HbA1c, RBC distribution width, LDL cholesterol, WBC count, lymphocyte count, monocyte count, HLSRC, and MCH.

### *Gold-standard genes*

For 9 binary traits and 41 quantitative traits considered in our analyses, we identified 49 expert-curated gold-standard genes with high confidence for CAD, Skin cancer, HDL cholesterol, Cholesterol, MRV, Calcium, LDL cholesterol and Platelet [13]. Two hundred one effector genes are identified in [51] for 43 binary and quantitative traits (Additional file 2: Supplementary Table 59).

### *Positive genes in Forgetta et al. [58]*

The positive genes for 12 traits considered in Forgetta et al. [58] were selected based on Mendelian disease genes or positive control drug targets. There are in total 494 positive genes across 12 diseases and traits, with 381 known to cause Mendelian forms of the disease and 113 drug targets. We focus on 208 gene-trait associations for 8 traits considered in our paper (type 2 diabetes, hyperthyroidism, BP-systolic, BP-diastolic, LDL-cholesterol, calcium, direct bilirubin, and red blood cell count).

### Computation cost

The estimated run time and memory use of BIGKnock depends on the sample size, gene length, number of variants in each gene and its corresponding regulatory elements, and the number of multiple knockoffs. For conducting BIGKnock gene-based tests on the UK Biobank data with 405,296 individuals and $M = 5$ multiple knockoffs, one trait requires 1817 CPU hours and on average 24 GB memory (for genes longer than 400kb, it requires >40 GB memory).

### Genome build

All genomic coordinates are given in GRCh37/hg19.

## Supplementary Information

---

**Additional file 1: Figures S1-22 and Table S1-5.**

**Additional file 2: Supplementary Tables 6-64.**

**Additional file 3.** Review history.

---

**Acknowledgements**
This research has been conducted using the UK Biobank Resource under Application Number 41849.

**Peer review information**
Stephanie McClelland and Anahita Bishop were the primary editors of this article and managed its editorial process and peer review in collaboration with the rest of the editorial team.

**Review history**
The review history is available as Additional file 3.

**Authors' contributions**
S.M. and I.I.-L. developed the concepts for the manuscript and proposed the method. S.M., C.W., A.K., L.L., Z.H., K.K., and I.I.-L. designed the analyses and applications and discussed results. S.M. and C.W. conducted the analyses. J.D. and K.K. helped interpret the results. S.M. and I.I.-L. prepared the manuscript and all authors contributed to editing the paper. The authors read and approved the final manuscript.

**Funding**
This work was supported by National Institute of Mental Health (NIMH) award MH095797 and National Institute on Aging (NIA) award AG072272.

**Availability of data and materials**
The UK Biobank data are available through formal application at http://www.ukbiobank.ac.uk. The summary statistics from SAIGE are downloaded at https://pheweb.org/UKB-SAIGE/ by inputting the phenotype or phecode [10] . The summary statistics from Neale lab UKB GWAS round 2 results are downloaded at http://www.nealelab.is/uk-biobank/. Gold-standard genes are available at https://github.com/opentargets/genetics-gold-standards [13]. All supplementary tables are also available at https://doi.org/10.5281/zenodo.7524304 [78].
We have implemented BIGKnock in a computationally efficient R package that can be applied generally to the analysis of other large biobank datasets. The package can be accessed at https://github.com/Iuliana-Ionita-Laza/BIGKnock [79], which is licensed under GPLv3. The source codes are also available at Zenodo https://doi.org/10.5281/zenodo.7524304 [78]. Software for SAIGE/SAIGE-Gene is available at https://github.com/weizhouUMICH/SAIGE [10, 11]. For the computational cost of fitting null GLMM, please refer to the SAIGE/SAIGE-Gene package [11].

## Declarations

### Ethics approval and consent to participate
Ethics approval was not needed for this study.

### Consent for publication
Not applicable.

### Competing interests
The authors declare that they have no competing financial interest.

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

## 

