## [**Additional file 3.** Review history. · Genome Biology]

Review History

First round of review

Reviewer 1

Were you able to assess all statistics in the manuscript, including the appropriateness of statistical tests used? Yes, and I have assessed the statistics in my report.

Comments to author:

Summary

Ma et al. propose a new gene-based test, BIGKnock, that incorporates functional data and "knockoffs" for conditional tests. BIGKnock extends a previous method that conducts gene-based, rare/common variant burden and variance tests which get aggregated using a Cauchy combination method. The novel extension here incorporates a computational approximation to efficiently generate knockoffs that are used to control Type I error. The method is applied to the UK Biobank where it is shown to identify fewer genes but more plausible genes.

Overall, this is a very interesting and useful method and a rigorously conducted analysis. My primary concern is that it was very difficult to distinguish the novelty in this approach versus the prior GeneScan3DKnock method, and some of the benchmarking should include GeneScan3DKnock to help clarify the similarities/differences. I also had some questions regarding potential overfitting and model assumptions.

Major comments

1. It is difficult to understand what is new methodologically in this work versus ref.6 (Ma et al. 2021 PNAS). The overview of methods claims "BIGKnock extends GeneScan3D by implementing the knockoff framework" and knockoffs are repeatedly emphasized but knockoffs were previously implemented using GeneScan3DKnock in ref.6. GeneScan3DKnock is not mentioned in this work at all. The rest of the overview of methods describes generating knockoffs, the τ computation, and the q-value computation which, from what I can tell, are identical to ref.6. It appears as though the primary innovation is the Shrinkage Leveraging algorithm for efficient knockoff computation using importance sampling, which is actually not mentioned in the overview of methods. If I'm understanding correctly, the most relevant comparison is BIGKnock versus GeneScan3DKnock, which should show that the methods produce equivalent results but with a lower computational cost for BIGKnock. Please provide this comparison. Additionally, the existence of GeneScan3DKnock needs to be mentioned, the Shrinkage Leveraging algorithm should be described in the overview of methods, and the innovation of BIGKnock versus prior work clearly defined. If GeneScan3DKnock was already shown to outperform GeneScan3D in prior work it is not clear why GeneScan3D is being used as the baseline here at all.
2. BIGKnock assumes a multivariate normal model of LD, but does this model still hold for the rare or low-frequency variants being tested? Please provide some assessment that the knockoffs approach (and particularly the SL algorithmic approximation) is still effective across all frequency ranges.
3. BIGKnock finds many fewer significant genes than GeneScan3D but this is presented as a positive finding ("This reduction in the number of significant associations is due to LD adjustment."). Can the authors show definitively that the reduction is due to LD adjustment

and not lower sensitivity for BIGKnock? If this is hard to demonstrate in real data with no ground truth, can they devise a simulation that would be convincing? If this cannot be shown definitively in real data or simulations, I think the quoted statement should be rephrased as a hypothesis rather than a conclusion. It would also help to clarify earlier in the section that fewer genes is considered better.

4. BIGKnock tends to identify genes that are closer to the lead GWAS SNP, could this in part be overfitting by including the GWAS SNP itself (or proxies) in the BIGKnock analysis? In particular, it would be helpful to justify (either mathematically or through simulations) that in a locus where there is a strong common GWAS association and a completely independent, weaker, rare burden association that BIGKnock does not lose sensitivity to detect the weaker independent association. Likewise, the contamination of rare variant associations with nearby common variant associations is a major confounder in such studies and it would be helpful to understand to what extent BIGKnock will identify false positive rare gene associations when there is a nearby common causal variant.

5. It would be helpful to provide a more complete explanation of what the knockoffs approach intends to adjust for in the Overview of BIGKnock. What are the assumptions and why is it the right choice for this task? In particular, it is claimed that the knockoffs provide an "LD adjustment" or "conditioning on nearby variants" but what is the model and what is the adjustment? In the absence of LD, does BIGKnock become identical to GeneScan3D? If two genes are perfectly correlated an *both* causal will BIGKnock infer that only one of them is significant or both or neither?

6. In the comparison with Backman et al. it is shown that BIGKnock achieves comparable results to conditioning on the lead GWAS SNP. But it's not clear what the downside is to conditioning on the lead GWAS SNP? What is the advantage of the BIGKnock approach here?

7. BIGKnock achieves precision of 0.66 and recall of 0.77 ($F1 = 0.71$), whereas closest gene achieves a precision of 0.61 and a recall of 0.85 ($F1 = 0.71$). It appears as though BIGKnock performs similarly to just taking the nearest gene? Please clarify which method is best.

8. If I understand correctly, the comparison between BIGKnock and cS2G or L2G is not entirely fair because BIGKnock cannot provide an estimate for traits without exome + GWAS data, whereas the other two methods do not require exome + GWAS data (i.e. they can make predictions for any SNP). If BIGKnock can make predictions about traits it hasn't seen this should be benchmarked directly, otherwise please clarify.

9. The methods describe analysis of burden of rare and common variants but the UKBiobank data section does not describe rare variants. Were rare variants from the UKBiobank actually used in the analysis? Please clarify.

Minor comments

The section starting on line 54 provides a paragraph-long list of quantitative traits and abbreviations, I recommend this be moved to a table.

Methods: The term "SKAT test" is used without a definition or citation, please formally define both "burden test" and "SKAT test" upon first use, ideally in terms of the equation on line 32.

Methods: Rather than diving into the implementation of the SL algorithm, it would help to first describe it conceptually and provide motivation.

Methods: "Note that co-regulation is still a problem and cannot be addressed by the proposed approach." This is a critical point and I recommend more formally defining co-regulation and how to interpret the BIGKnock results when there is co-regulation.

Reviewer 2

Were you able to assess all statistics in the manuscript, including the appropriateness of statistical tests used? Yes, and I have assessed the statistics in my report.

Comments to author:

Review: BIGKnock: Fine-mapping gene-based associations via knockoff analysis of biobank-scale data with applications to UK Biobank

Overview: Ma et al present BIGKnock, a statistical framework that integrates chromatin interaction data together with knock-off based associations from biobank scale GWAS data. The presented approach builds on previous gene-based knock-off work from this group (GeneScan3D), but leverages recent computational developments to scale to the extremely-large data setting of biobank data (e.g., UK Biobank). They demonstrate their approach by applying it to several disease and quantitative traits from the UK Biobank alongside their previous method. They also investigate its performance relative to newer gene mapping methods (S2G, L2G). I found the manuscript presentation to be a bit tedious and lacking polish. Additionally, and more importantly, I have some concerns regarding the lack of clarity in what is novel in BIGKnock versus GeneScan3D, and the lack of simulations. I detail my thoughts below.

Major comments:

1. The authors describe BIGKnock as a gene/burden-based framework that extends their earlier approach GeneScan3D, but implements a knock-off framework for hypothesis testing in an FDR setting. The original GeneScan3D paper describes not only the gene/burden-based framework, but also provides a knock-off framework (GeneScan3DKnock) that reflects much of the important statistical details provided here, alongside an application to UK Biobank data. What is novel, as the authors note to some extent in the introduction, but fail to sufficiently emphasize in the results/overview, is the incorporation of the shrinkage-leveraging algorithm in the knock-off generation.

Given this, I feel it would be much more appropriate to compare BIGKnock versus the GeneScan3DKnock to demonstrate any novel findings. This would provide much needed information on whether or not the shrinkage implementation, which clusters genetic variants to reduce the total number of predictors, affects the stability of identified genes.

2. The authors previous work provided simulations to assess power and FDR calibration of GeneScan3DKnock, but I see no such simulations provided here for BIGKnock. Having simulations to assess the statistical stability of BIGKnock's incorporation of the speedup would greatly help with interpreting results.

3. Given the novelty of this work is the speedup I would have expected to see discussion or presentation of results regarding the improved runtime or memory usage, but could not find any details beyond those presented on lines 28-36 of page 18. So I am left unsure as to how important the shrinkage/clustering approach is to not only the statistical results as compared

with GeneScan3DKnock, but also whether or not it actually improves runtime/memory usage.

We would like to thank the two reviewers for their helpful comments. We have revised the paper and added additional analyses to show the statistical performance of BIGKnock in terms of FDR and power, and also the computational efficiency gained in BIGKnock over the previously proposed gene-based test GeneScan3DKnock. All the changes are marked in blue font. Below please find point-by-point responses to the reviewers' comments.

Reviewer #1:

Summary

Ma et al. propose a new gene-based test, BIGKnock, that incorporates functional data and "knockoffs" for conditional tests. BIGKnock extends a previous method that conducts gene-based, rare/common variant burden and variance tests which get aggregated using a Cauchy combination method. The novel extension here incorporates a computational approximation to efficiently generate knockoffs that are used to control Type I error. The method is applied to the UK Biobank where it is shown to identify fewer genes but more plausible genes.

Overall, this is a very interesting and useful method and a rigorously conducted analysis. My primary concern is that it was very difficult to distinguish the novelty in this approach versus the prior GeneScan3DKnock method, and some of the benchmarking should include GeneScan3DKnock to help clarify the similarities/differences. I also had some questions regarding potential overfitting and model assumptions.

Response: Thank you for the nice summary. The main improvement over GeneScan3DKnock is indeed the computational efficiency. The knockoff generation is very computationally intensive and, especially at the scale of UKBB datasets, the previous method (GeneScan3DKnock) cannot be applied. Therefore, we propose BIGKnock that employs a shrinkage leveraging algorithm to improve computational efficiency. Our new simulations demonstrate these features (see Figure 1 in the manuscript and below). Another improvement is that we now implement GLMM (instead of GLM) to handle sample relatedness in such datasets.

The applications to the UK Biobank are also of independent interest as they essentially provide new insights for the studied traits. In the UKBB we chose to compare with the conventional version of the gene-based test to particularly highlight the benefit the knockoff test (in terms of prioritization of likely causal genes) over the commonly used tests; also note that scalability to UKBB size is a primary motivation for developing our test, and it is infeasible to run our previous test GeneScan3DKnock on the UKBB data.

Figure 1: Power, FDR and computing time comparisons for different methods. (a) and (b) Power and FDR comparisons between GeneScan3D-BH, GeneScan3DKnock and BIGKnock (M=5 knockoffs) for continuous and binary traits. (c) Computing time for different methods to generate knockoffs: GeneScan3DKnock and BIGKnock (with shrinkage algorithmic leveraging). The computing time were evaluated based on a gene with 1,347 variants, varying the sample size from 1,000 to 400,000.

Major comments

1. It is difficult to understand what is new methodologically in this work versus ref.6 (Ma et al. 2021 PNAS). The overview of methods claims "BIGKnock extends GeneScan3D by implementing the knockoff framework" and knockoffs are repeatedly emphasized but knockoffs were previously implemented using GeneScan3DKnock in ref.6. GeneScan3DKnock is not mentioned in this work at all. The rest of the overview of methods describes generating knockoffs, the τ computation, and the q -value computation which, from what I can tell, are identical to ref.6. It appears as though the primary innovation is the Shrinkage Leveraging algorithm for efficient knockoff computation using importance sampling, which is actually not mentioned in the overview of methods. If I'm understanding correctly, the most relevant comparison is BIGKnock versus GeneScan3DKnock, which should show that the methods produce equivalent results but with a lower computational cost for BIGKnock. Please provide this comparison. Additionally, the existence of GeneScan3DKnock needs to be mentioned, the Shrinkage Leveraging algorithm should be described in the overview of methods, and the innovation of BIGKnock versus prior work

clearly defined. If GeneScan3DKnock was already shown to outperform GeneScan3D in prior work it is not clear why GeneScan3D is being used as the baseline here at all.

Response: Thank you for the comment, and we apologize for the confusion. We have revised the manuscript to highlight the difference with our previously proposed method. As mentioned above, the main contribution is to make the GeneScan3DKnock scalable to large biobank datasets with hundreds of thousands of individuals and millions of variants. We clarify this aspect in this revision. We also provide new simulations to show the computational improvement. We still think that our baseline for comparison that is of interest to other researchers in the UKBB is with GeneScan3D because that corresponds to the conventional test that is commonly used by the community (i.e. no knockoffs) so we want to highlight the proposed method's advantages relative to the standard. In addition, we show that in terms of power and FDR, BIGKnock and GeneScan3DKnock are very similar. In fact, GeneScan3DKnock cannot even be performed at the UKBB scale in a computationally efficient way, so we could not provide a comparison in the real data applications.

2. BIGKnock assumes a multivariate normal model of LD, but does this model still hold for the rare or low-frequency variants being tested? Please provide some assessment that the knockoffs approach (and particularly the SL algorithmic approximation) is still effective across all frequency ranges.

Response: The knockoff generator is based on the observation that the correlation among genetic variants approximately exhibits a block diagonal structure due to the nature of linkage disequilibrium in a genetic region. The genotypes are modeled as dosage values (i.e. continuous-value genotypes generated by imputation methods) and we assume a multivariate normal distribution with block diagonal covariance matrix. Although it is true that this is an approximation, it seems to perform well in practice as long as the LD is well captured. Our focus for this paper has been on common variants, and we leave the extension to rare variants for future work (we clarify this aspect in the manuscript). It is true that for rare variants this approximation may not work that well.

3. BIGKnock finds many fewer significant genes than GeneScan3D but this is presented as a positive finding ("This reduction in the number of significant associations is due to LD adjustment."). Can the authors show definitively that the reduction is due to LD adjustment and not lower sensitivity for BIGKnock? If this is hard to demonstrate in real data with no ground truth, can they devise a simulation that would be convincing? If this cannot be shown definitively in real data or simulations, I think the quoted statement should be rephrased as a hypothesis rather than a conclusion. It would also help to clarify earlier in the section that fewer genes is considered better.

Response: Indeed, the main goal is to reduce the number of false associations (those due to LD) at a locus. We have revised the introduction to emphasize this point. Because the knockoff-based approach performs conditional testing (conditional on LD) we expect to see reductions in number of associations due to LD. It is the same principle as in fine-mapping studies for GWAS loci where

we can reduce the number of false associations we obtain in marginal testing due to LD. Also, we can see from Figure 1 that GeneScan3D has increased FDR, while the knockoff-based methods can control FDR which clearly is related to the ability of the knockoff tests to reduce false positives due to LD.

We have also added a figure to more clearly show evidence that the reduced number of associations is not due to reduced sensitivity. In particular we show that when we look at likely causal genes identified using rare variant loss-of-function variants in exome sequencing studies and gold-standard genes (Mendelian disease genes and drug targets), the knockoff-based test discovers 89%-98% of those genes, compared to ~63% of the genes significant under conventional tests, suggesting that the knockoff-based test is removing false associations at a locus (due to LD since LD is the main confounding factor) but tends to keep the true causal associations.

Figure 2: **BIGKnock retention rates for all genes, Backman effector genes and the gold-standard genes, with two-sided p-values.** The retention rates are computed as the proportions of BIGKnock significant genes among the GeneScan3D significant genes using three different gene sets: all genes and two gene sets enriched for causal genes (Backman effector genes, and gold-standard genes).

Finally, we show below a simulated example replicating the UKBB analysis in Figure 4b. This example is only meant to illustrate what we believe is happening in Figure 4b. Specifically, we perform simulations at the ASGR1 locus. There are 55 genes at this locus and we set 10% of the variants in the gene+buffer regions of ASGR1 and CD68 to be causal, all located within randomly selected 10-Kb windows. The effect sizes are $\beta_k = c|\log_{10} m_k|$, where m_k is the MAF for the k -

th causal variant and $c = 0.25$. Other settings are similar to the simulation studies on page 3 of the manuscript. The replicate in Figure 3 below shows that 27 genes are significant under GeneScan3D-BH with $q\text{-value} < 0.1$, while only the two assumed causal genes are significant under BIGKnock (ASGR1 and CD68). Therefore, BIGKnock greatly reduces the number of false positive associations.

Figure 3: **Simulation example at the ASGR1 locus.** BIGKnock identifies two causal genes (ASGR1 and CD68), while GeneScan3D-BH identifies 27 significant genes with false associations due to LD at this locus.

4. *BIGKnock tends to identify genes that are closer to the lead GWAS SNP, could this in part be overfitting by including the GWAS SNP itself (or proxies) in the BIGKnock analysis? In particular, it would be helpful to justify (either mathematically or through simulations) that in a locus where there is a strong common GWAS association and a completely independent, weaker, rare burden association that BIGKnock does not lose sensitivity to detect the weaker independent association. Likewise, the contamination of rare variant associations with nearby common variant associations is a major confounder in such studies and it would be helpful to understand to what extent BIGKnock will identify false positive rare gene associations when there is a nearby common causal variant.*

Response: The knockoff generation is done independently of the phenotype, so there should be no bias from the knockoff generation towards the top significant SNP.

If two loci are completely independent, then the detection of one locus will not be affected by the other locus. This can be understood from the fact that the knockoff performs conditional testing, but if the loci are independent then the conditioning has no effect. We note that our

paper has focused on detecting association at the gene level; we do not perform testing at the variant level and therefore the reviewer's question is not directly applicable to this scenario. We have however done an experiment in KnockoffScreen (PMID: 34035245) to address exactly this issue, and have shown that "the conventional tests tend to identify a large number of false positives due to the shadow effect. In contrast, KnockoffScreen has a significantly reduced number of false positives, demonstrating that it is able to distinguish the effect of rare variants from that of common variants nearby."

*5. It would be helpful to provide a more complete explanation of what the knockoffs approach intends to adjust for in the Overview of BIGKnock. What are the assumptions and why is it the right choice for this task? In particular, it is claimed that the knockoffs provide an "LD adjustment" or "conditioning on nearby variants" but what is the model and what is the adjustment? In the absence of LD, does BIGKnock become identical to GeneScan3D? If two genes are perfectly correlated and *both* causal will BIGKnock infer that only one of them is significant or both or neither?*

Response: Knockoff is a statistical framework to do variable selection in the presence of correlations. We have added more details on the overall goal of knockoff framework. The adjustment for LD is due to the way the knockoff variables are generated, conditional on the genetic variants in the region. So the null hypothesis tested by the knockoff is a conditional null hypothesis: $H_0: Y \perp G_j \mid \mathbf{G}_{-j}$ for $j = 1, \dots, p$.

It would be quite hard for two genes to be completely correlated, because genes include many variants in gene body and regulatory elements, so the correlation will tend to be weaker than say among two variants. For variants, in general, the knockoff framework cannot separate them if they are completely correlated. Therefore they will be detected as a group.

6. In the comparison with Backman et al. it is shown that BIGKnock achieves comparable results to conditioning on the lead GWAS SNP. But it's not clear what the downside is to conditioning on the lead GWAS SNP? What is the advantage of the BIGKnock approach here?

Response: This comparison with Backman et al. is to show that the knockoff analysis discovers many of the likely causal genes identified in Backman et al.. Backman et al focused on rare pLOF variants and performed gene-based tests with these variants. Although this is a better approach to identify causal genes, it requires large sequencing studies. Our approach is using GWAS data and therefore attempts to identify causal genes by LD adjustment.

Knockoff construction is different from the conditioning on lead SNP. In the knockoff construction the phenotype is not being used, and there is no conditioning on the top SNP. Conditioning on the top SNP is a simple approach that assumes that the lead SNP is causal and then tries to identify independent signals. However, we want to clarify that the goal in this particular analysis is to highlight the ability of the knockoff to keep (not remove) likely causal genes.

7. BIGKnock achieves precision of 0.66 and recall of 0.77 (F1 = 0.71), whereas closest gene achieves a precision of 0.61 and a recall of 0.85 (F1 = 0.71). It appears as though BIGKnock performs similarly to just taking the nearest gene? Please clarify which method is best.

Response: This comparison is limited because we do not have good test datasets for causal genes, and only meant to show that BIGKnock can compete with state-of-the-art SNP to gene strategies. Although one can think of BIGKnock as a gene prioritization tool at GWAS loci, BIGKnock is first of all a formal gene-based test that produces a q value. Many of the known causal genes available are biased towards the 'nearest gene', therefore the nearest gene approach seems to perform very well. But that is a limitation to the data we have at this moment.

8. If I understand correctly, the comparison between BIGKnock and cS2G or L2G is not entirely fair because BIGKnock cannot provide an estimate for traits without exome + GWAS data, whereas the other two methods do not require exome + GWAS data (i.e. they can make predictions for any SNP). If BIGKnock can make predictions about traits it hasn't seen this should be benchmarked directly, otherwise please clarify.

Response: Yes, we have clarified this point. BIGKnock is a gene-based test in the first place (note however that only GWAS data is enough for BIGKnock – there is no need for exome data); that comparison is meant to illustrate its performance in the context of gene prioritization at GWAS loci. That analysis also illustrates the complementary nature of the different approaches and how combining them can lead to better performance.

9. The methods describe analysis of burden of rare and common variants but the UKBiobank data section does not describe rare variants. Were rare variants from the UKBiobank actually used in the analysis? Please clarify.

Response: We have clarified that our current analyses have focused on common variants. Although rare variants can be included, generating valid knockoff for rare variants is more difficult, so we leave this as future work for more thorough investigation. In the UKBB we used $MAF > 0.01$.

Minor comments

The section starting on line 54 provides a paragraph-long list of quantitative traits and abbreviations, I recommend this be moved to a table.

Response: We have done that (see Table S2).

Methods: The term "SKAT test" is used without a definition or citation, please formally define both "burden test" and "SKAT test" upon first use, ideally in terms of the equation on line 32.

Response: The Burden and SKAT tests are two types of gene-based tests used in the literature. We have added to citations to the original paper (page 13) and they are also described on page 16 in more detail.

Methods: Rather than diving into the implementation of the SL algorithm, it would help to first describe it conceptually and provide motivation.

Response: Thank you for the suggestion. We have added more intuitive description on page 13.

Methods: "Note that co-regulation is still a problem and cannot be addressed by the proposed approach." This is a critical point and I recommend more formally defining co-regulation and how to interpret the BIGKnock results when there is co-regulation.

Response: Co-regulation (in our narrow interpretation) means that a causal enhancer may regulate multiple genes. Then that means that the same enhancer is included in the definition of multiple genes so there is overlap between genes. That type of scenario (overlap of causal enhancers) cannot be disentangled using the proposed approach. We have clarified this part in the manuscript.

Reviewer #2:

Overview: Ma et al present BIGKnock, a statistical framework that integrates chromatin interaction data together with knock-off based associations from biobank scale GWAS data. The presented approach builds on previous gene-based knock-off work from this group (GeneScan3D), but leverages recent computational developments to scale to the extremely-large data setting of biobank data (e.g., UK Biobank). They demonstrate their approach by applying it to several disease and quantitative traits from the UK Biobank alongside their previous method. They also investigate its performance relative to newer gene mapping methods (S2G, L2G). I found the manuscript presentation to be a bit tedious and lacking polish. Additionally, and more importantly, I have some concerns regarding the lack of clarity in what is novel in BIGKnock versus GeneScan3D, and the lack of simulations. I detail my thoughts below.

Response: Thank you for your comment. We have revised the presentation to improve the description, highlighted the novel aspects and added simulations to show the empirical performance.

Major comments:

1. The authors describe BIGKnock as a gene/burden-based framework that extends their earlier approach GeneScan3D, but implements a knock-off framework for hypothesis testing in an FDR setting. The original GeneScan3D paper describes not only the gene/burden-based framework, but also provides a knock-off framework (GeneScan3DKnock) that reflects much of the important statistical details provided here, alongside an application to UK Biobank data. What is novel, as the authors note to some extent in the introduction, but fail to sufficiently emphasize in the results/overview, is the incorporation of the shrinkage-leveraging algorithm in the knock-off generation.

Given this, I feel it would be much more appropriate to compare BIGKnock versus the GeneScan3DKnock to demonstrate any novel findings. This would provide much needed information on whether or not the shrinkage implementation, which clusters genetic variants to reduce the total number of predictors, affects the stability of identified genes.

Response: Thank you for your comment. The main improvement over GeneScan3DKnock is indeed the computational efficiency. The knockoff generation is very computationally intensive and, especially at the scale of UKBB datasets, the previous method (GeneScan3DKnock) cannot be applied. Therefore, we propose BIGKnock that employs a shrinkage leveraging algorithm to improve computational efficiency. Our new simulations demonstrate these features (see Figure 1 in the manuscript and below). Another improvement is that we now implement GLMM (instead of GLM) to handle sample relatedness in such datasets.

We still think that our baseline for comparison that is of interest to other researchers in the UKBB application is with GeneScan3D because that corresponds to the conventional test that is commonly used by the community (i.e. no knockoffs) so we want to highlight the proposed method's advantages relative to the standard. In addition, we show that in terms of power and FDR, BIGKnock and GeneScan3DKnock are very similar. In fact, GeneScan3DKnock cannot even be performed at the UKBB scale in a computationally efficient way, so we could not provide a comparison in the real data applications.

The applications to the UK Biobank are also of independent interest as they essentially provide new insights for the studied traits. In the UKBB we chose to compare with the conventional version of the gene-based test to particularly highlight the benefit the knockoff test (in terms of prioritization of likely causal genes) over the commonly used tests.

Figure 1: **Power, FDR and computing time comparisons for different methods.** (a) and (b) Power and FDR comparisons between GeneScan3D-BH, GeneScan3DKnock and BIGKnock (M=5 knockoffs) for continuous and binary traits. (c) Computing time for different methods to generate knockoffs: GeneScan3DKnock and BIGKnock (with shrinkage algorithmic leveraging). The computing time were evaluated based on a gene with 1,347 variants, varying the sample size from 1,000 to 400,000.

2. *The authors previous work provided simulations to assess power and FDR calibration of GeneScan3DKnock, but I see no such simulations provided here for BIGKnock. Having simulations*

to assess the statistical stability of BIGKnock's incorporation of the speedup would greatly help with interpreting results.

Response: We apologize for that omission and now have provided additional simulations (see previous response).

3. Given the novelty of this work is the speedup I would have expected to see discussion or presentation of results regarding the improved runtime or memory usage, but could not find any details beyond those presented on lines 28-36 of page 18. So I am left unsure as to how important the shrinkage/clustering approach is to not only the statistical results as compared with GeneScan3DKnock, but also whether or not it actually improves runtime/memory usage.

Response: We have provided additional simulations and also emphasize more the contribution of this manuscript in allowing previously developed test GeneScan3DKnock to be scalable to biobank sized datasets such as illustrated here with the UKBB application.

Second round of review

Reviewer 1

The authors have addressed all of my concerns and the manuscript is much improved.

Reviewer 2

I have no additional comments at this time